# AdaWorld: Learning Adaptable World Models with Latent Actions

**Shenyuan Gao**[1]  **Siyuan Zhou**[1]  **Yilun Du**[2]  **Jun Zhang**[1]  **Chuang Gan**[3][4]

adaptable-world-model.github.io

## Abstract

World models aim to learn action-controlled future prediction and have proven essential for the development of intelligent agents. However, most existing world models rely heavily on substantial action-labeled data and costly training, making it challenging to adapt to novel environments with heterogeneous actions through limited interactions. This limitation can hinder their applicability across broader domains. To overcome this limitation, we propose *AdaWorld*, an innovative world model learning approach that enables efficient adaptation. The key idea is to incorporate action information during the pretraining of world models. This is achieved by extracting latent actions from videos in a self-supervised manner, capturing the most critical transitions between frames. We then develop an autoregressive world model that conditions on these latent actions. This learning paradigm enables highly adaptable world models, facilitating efficient transfer and learning of new actions even with limited interactions and finetuning. Our comprehensive experiments across multiple environments demonstrate that AdaWorld achieves superior performance in both simulation quality and visual planning.

## 1. Introduction

Intelligent agents should perform effectively across various tasks (Reed et al., 2022; Lee et al., 2022; Durante et al., 2024; Raad et al., 2024). A promising solution to this objective is developing world models that can simulate different environments (Wu et al., 2023; 2024; Yang et al., 2024c; Hansen et al., 2024). Recent world models are typically initialized from pretrained video models (Xiang et al., 2024; Agarwal et al., 2025; He et al., 2025). Despite im-

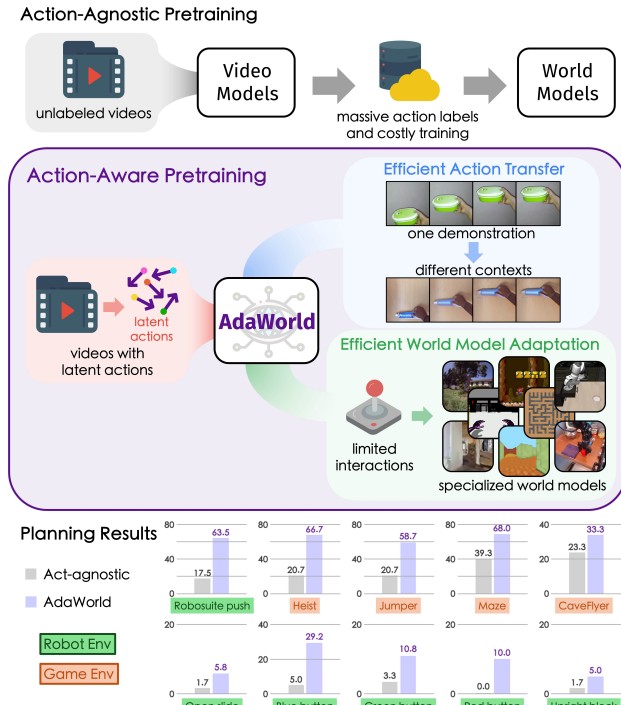

*Figure 1.* **Different world model learning paradigms.** Prior methods often require expensive labeling and training to achieve action controllability in new environments. To overcome this, we introduce latent actions as a unified condition for action-aware pretraining from videos, enabling highly adaptable world modeling. Our world model, dubbed *AdaWorld*, can readily transfer actions across contexts without training. By initializing the control interface with the corresponding latent actions, AdaWorld can also be adapted into specialized world models efficiently and achieve significantly better planning results than the action-agnostic baseline.

proved generalization, these models still require substantial action labels and high training costs to acquire precise action controllability. While pseudo labels can be annotated for videos (Baker et al., 2022; Zhang et al., 2022), defining a unified action format for general environments is challenging. As a result, existing methods often require costly training when adapting to new environments with varying action specifications (Gao et al., 2024; Chi et al., 2024; Che et al., 2025). These limitations pose great challenges for transferring and learning new actions based on limited

[1]HKUST [2]Harvard [3]UMass Amherst [4]MIT-IBM Watson AI Lab. Primary contact to Shenyuan Gao <sygao@connect.ust.hk>.

*Proceedings of the $42^{st}$ International Conference on Machine Learning*, Vancouver, Canada. PMLR 267, 2025. Copyright 2025 by the author(s).

interactions and finetuning.

As humans, we can estimate the effects of different actions through limited experiences (Ha & Schmidhuber, 2018). This ability likely arises from our internal representations of actions learned from extensive observations (Rizzolatti et al., 1996; Romo et al., 2004; Dominici et al., 2011). These common knowledge can be reused across different contexts and associated with specific action spaces efficiently (Rybkin et al., 2019; Schmeckpeper et al., 2020; Sun et al., 2024). Consequently, humans can easily transfer observed actions to various contexts and imagine the transitions of new environments through a few interactions (Poggio & Bizzi, 2004). These insights motivate us to ponder: can we achieve human-like adaptability in world modeling by learning transferrable action representations from observations?

In this paper, we propose *AdaWorld*, an innovative pretraining approach for highly adaptable world models. Unlike previous methods that only pretrain on action-agnostic videos, we argue that incorporating action information during pretraining will significantly enhance the adaptability of world models. As illustrated in Figure 1, the adaptability of AdaWorld primarily manifests in two aspects: (1) Given one demonstration of an action, AdaWorld can readily transfer that action to various contexts without further training. (2) It can also be efficiently adapted into specialized world models with raw action inputs via minor interactions and finetuning, enabling more effective planning in various environments.

AdaWorld consists of two key components: a latent action autoencoder that extracts actions from unlabeled videos, and an autoregressive world model that takes the extracted actions as conditions. Our main challenge is that in-the-wild videos often involve complicated contexts (*e.g.*, colors and textures), which hinders effective action recognition. To overcome this challenge, we introduce an information bottleneck design to our latent action autoencoder. Specifically, the latent action encoder extracts a compact encoding from two consecutive frames. We refer to this encoding as *latent action* hereinafter, as it is used to represent the transition between these two frames. Based on the latent action and the former frame, the latent action decoder makes its best effort to predict the subsequent frame. By minimizing the prediction loss using the minimal information encoded in the latent action, our autoencoder is encouraged to disentangle the most critical action from its context. Unlike previous methods (Bruce et al., 2024; Chen et al., 2024b; Ye et al., 2025) that focus on playability and behavior cloning, we compress the latent actions into a continuous latent space to maximize expressiveness and enable flexible composition. We find that our latent actions are context-invariant and can be effectively transferred across different contexts.

We then pretrain an autoregressive world model that conditions on the latent actions. Thanks to the strong transferabil-

ity of our latent actions, the resulting world model is readily adaptable to various environments. In particular, since our world model has learned to simulate different actions represented by any latent actions, adapting to a new environment is akin to finding the mapping of corresponding latent actions for its action space. Given one demonstration of an action, our model can readily transfer the demonstrated action by extracting the latent action with latent action encoder and reusing it across different contexts. When action labels are provided, we can similarly obtain their latent actions and initialize the control interface efficiently. This enables us to adapt our model to specialized world models with minimal finetuning. When only a limited number of interactions are available (*e.g.*, 50 interactions), our approach is significantly more efficient than pretraining from action-agnostic videos.

Note that our approach is as scalable as existing video pretraining methods (Seo et al., 2022; Mendonca et al., 2023; Wu et al., 2023; Agarwal et al., 2025; He et al., 2025; Yu et al., 2025). To enhance the generalization ability of AdaWorld, we collect a large corpus of videos from thousands of environments through automated generation. The resulting dataset encompasses extensive interactive scenarios, spanning from ego perspectives and third-person views to virtual games and real-world activities. After action-aware pretraining at scale, we show that the adaptability of AdaWorld can seamlessly generalize to a wide variety of domains.

In summary, we make the following contributions:

- We present AdaWorld, an autoregressive world model that is highly adaptable across various environments. It can readily transfer actions to different contexts and allows efficient adaptation with limited interactions.

- We establish AdaWorld on a large-scale dataset sourced from extremely diverse environments. After extensive pretraining, AdaWorld demonstrates strong generalization capabilities across various domains.

- We conduct comprehensive experiments across multiple environments to verify the efficacy of AdaWorld. Our model achieves promising results in action transfer, world model adaptation, and visual planning.

## 2. Method

In this section, we first introduce the architectural design of our latent action autoencoder (Sec. 2.1). By leveraging the latent actions as conditions, we then build an autoregressive world model through action-aware pretraining (Sec. 2.2). Finally, we demonstrate how our model facilitates highly adaptable world modeling (Sec. 2.3).

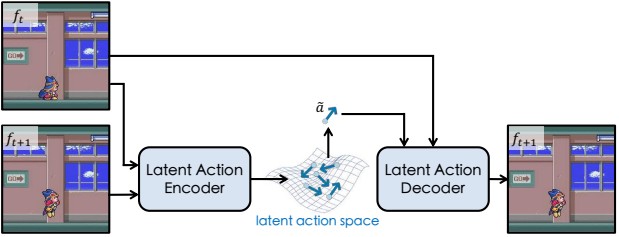

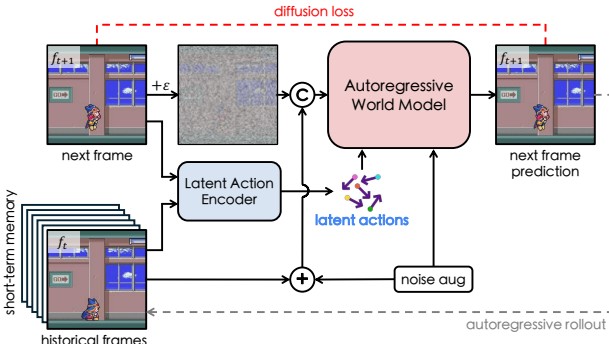

*Figure 2.* **Latent action autoencoder.** With an information bottleneck design, our latent action autoencoder is able to extract the most critical action information from videos and compresses it into a continuous latent action.

## 2.1. Latent Action Autoencoder

Instead of only taking actionless videos for world model pretraining, our key innovation is to incorporate action information during the pretraining phase. Benefiting from the pretrained knowledge of action controllability, the resulting world models can be efficiently adapted using limited ground truth actions. However, action labels are rarely available for in-the-wild videos. While a common practice is to collect these labels through interactions, collecting them across a multitude of environments incurs significant labor. Furthermore, defining a unified action format across diverse environments is often unfeasible, and current world models require costly training to accommodate new action formats.

To address these challenges, instead of relying on explicit action annotations, we propose extracting latent actions from videos as a unified condition for world model pretraining. Nevertheless, in general videos, the action information is often entangled with the contexts, posing significant difficulty for effective action recognition. Inspired by the observation that agents' actions often drive the dominant variation in most interactive scenarios (Rybkin et al., 2019; Menapace et al., 2021; 2022; Bruce et al., 2024), we introduce an information bottleneck to automatically differentiate actions from observations.

To be specific, we instantiate a latent action autoencoder based on the Transformer architecture (Vaswani et al., 2017), where the encoder extracts the latent action $\tilde{a}$ from two consecutive frames $f_{t:t+1}$, and the decoder predicts the subsequent frame $f_{t+1}$ based on the latent action $\tilde{a}$ and the former frame $f_t$. The latent action encoder divides two frames $f_{t:t+1}$ into image patches of size $16 \times 16$. These patches are then projected to patch embeddings and flattened along the spatial dimension. Afterwards, they are concatenated with two learnable tokens $a_{t:t+1}$. Sinusoidal position embeddings (Dosovitskiy et al., 2021) are also applied to each frame to indicate the spatial information. To efficiently encode the tokens from these two frames, we employ a spatiotemporal Transformer (Bruce et al., 2024) with $L$ stacked blocks. Each block comprises interleaved spatial and temporal attention modules, followed by a feed-forward network.

*Figure 3.* **Action-aware pretraining.** We extract latent actions from unlabeled videos using the latent action encoder. By leveraging the extracted actions as a unified condition, we pretrain a world model that can perform autoregressive rollouts at inference.

The spatial attention can attend to all tokens within each frame, while the temporal attention has access to the two tokens in the same spatial positions across the two frames. We also incorporate rotary embeddings (Su et al., 2024) in temporal attentions to indicate the causal relationship. After sufficient attention correlations, the learnable tokens $a_{t:t+1}$ can adaptively aggregate the temporal dynamics between the two input frames. We then discard all tokens and only project $a_{t+1}$ to estimate the posterior of the latent action $(\mu_{\tilde{a}}, \sigma_{\tilde{a}})$ following the standard VAE (Kingma & Welling, 2014). Subsequently, we sample $\tilde{a}$ from the approximated posterior and attach it to $f_t$, which is then sent to the latent action decoder. The latent action decoder is a spatial Transformer that predicts the subsequent frame $f_{t+1}$ in the pixel space. The whole latent action autoencoder is optimized with the VAE objective:

$$\mathcal{L}^{pred}_{\theta,\phi}(f_{t+1}) = \mathbb{E}_{q_\phi(\tilde{a}|f_{t:t+1})} \log p_\theta(f_{t+1}|\tilde{a}, f_t) \quad (1)$$
$$- D_{KL}(q_\phi(\tilde{a}|f_{t:t+1})||p(\tilde{a})).$$

Compared to the original pixel space, the dimension of our latent action is extremely compact. Hence, it is challenging to forward the entire subsequent frame to the decoder via latent action. To minimize the prediction error of the subsequent frame, the latent action $\tilde{a}$ must encapsulate the most critical variations relative to the former frame. This results in context-invariant action representations that closely correspond to the true actions taken by the agents.

Nevertheless, we empirically find that our latent action autoencoder, trained using the aforementioned formulation, struggles to express diverse transitions between frames. This problem arises because the standard VAE imposes a strong constraint on posterior distributions. Conversely, removing this constraint may compromise the disentanglement ability of VAE (Burgess et al., 2017). To remedy this, we adopt the $\beta$-VAE formulation (Higgins et al., 2017; Alemi et al.,

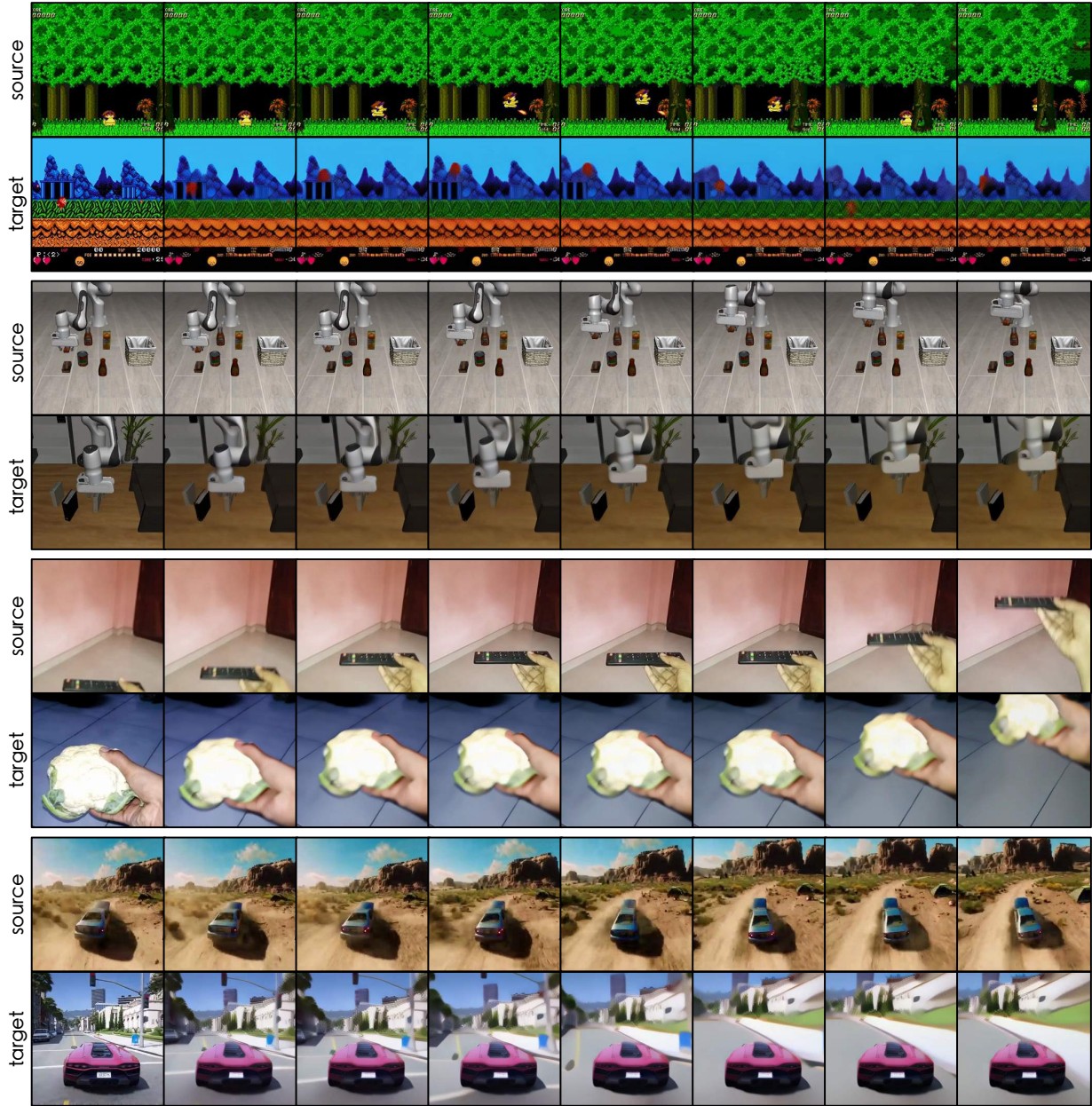

*Figure 4.* **Action transfer from demonstrations.** By extracting and reusing the latent actions in different contexts, AdaWorld can readily transfer the demonstrated actions from source videos to various target scenes without training. Please see Appendix C for more results.

2017), which introduces an adjustable hyperparameter $\beta$:

$$\mathcal{L}_{\theta,\phi}^{pred}(f_{t+1}) = \mathbb{E}_{q_\phi(\tilde{a}|f_{t:t+1})} \log p_\theta(f_{t+1}|\tilde{a}, f_t) \quad (2)$$
$$- \beta\, D_{KL}(q_\phi(\tilde{a}|f_{t:t+1})||p(\tilde{a})).$$

The additional hyperparameter enables us to flexibly control the information contained by the latent actions. In practice, we empirically adjust this hyperparameter to achieve a good trade-off between expressiveness and context disentangling ability of our latent actions. As shown in Figure 4, our latent action autoencoder can extract context-invariant actions that are transferrable across different contexts.

## 2.2. Action-Aware Pretraining

After training the latent action autoencoder, we can use its encoder to automatically extract action information from videos. This allows us to incorporate action information for world model pretraining, which we refer to as *action-aware pretraining*. To realize this, we pretrain a world model that predicts the next frame conditioned on the current latent action. As shown in Figure 3, we utilize the latent action encoder to extract the latent actions between frames and send them as inputs for our world model. Unlike previous methods that often predict video clips (Yang et al., 2024c;

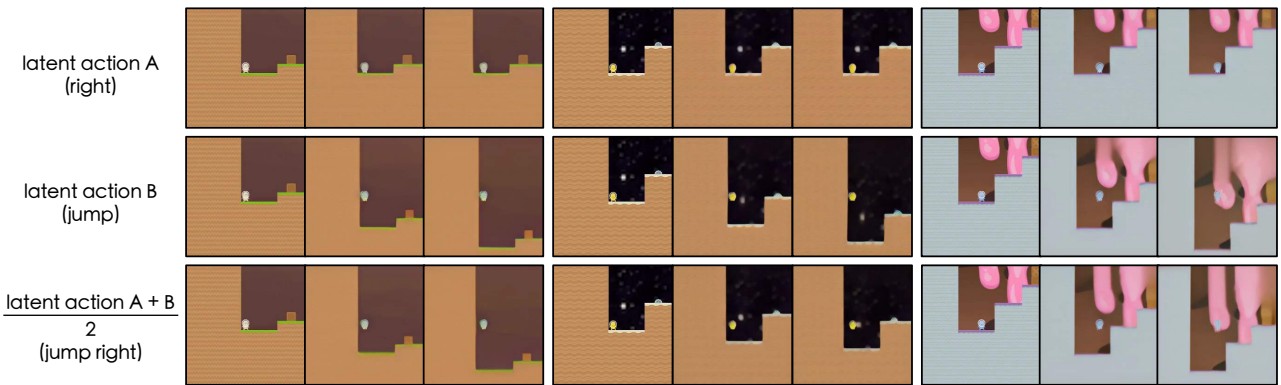

*Figure 5.* **Action composition.** We compose two actions by averaging their latent actions in the continuous latent space, resulting in a new action that merges the functions of both. This indicates that our latent action space is semantically continuous in the meanings of actions.

Xiang et al., 2024; Agarwal et al., 2025), our model supports frame-level control, offering finer granularity for interaction. To ensure smooth transitions, we maintain a short-term memory with $K$ historical frames. During inference, our model can predict a sequence of future frames by autoregressively repeating the next-frame prediction process and appending the predicted frames to the memory.

Although it may seem straightforward to repurpose the latent action decoder as the function of world model, it only makes coarse predictions via a single forward pass, resulting in significant quality degradation after several interactions. To achieve genuine predictions, we establish an independent world model based on diffusion models. Specifically, we initialize the world model with Stable Video Diffusion (SVD) (Blattmann et al., 2023), a latent diffusion model trained with the EDM framework (Karras et al., 2022). Different from the original SVD, we only denoise one noisy frame each time. To enable deep aggregation with the action information, the latent action is concatenated with both the timestep embedding and the CLIP image embedding from the original SVD. The last frame in memory is used as the condition image of SVD. To inherit the pretrained temporal modeling capability, we encode the historical frames using the SVD image encoder and concatenate them with the noise latent map of the frame to predict. Since the number of available historical frames may vary in practice, we randomly sample historical frames with a maximum length of 6 during training and send the memory length condition to the world model. Following previous practices (He et al., 2022; Valevski et al., 2025), noise augmentation is also applied to corrupt the historical frames during training. This augmentation can effectively alleviate the long-term drift problem, even when no noise is applied during inference. We pretrain the world model on our large-scale dataset by minimizing the following diffusion loss:

$$\mathcal{L}_{\text{pretrain}} = \mathbb{E}_{\boldsymbol{x}_0, \epsilon, t}\left[\|\boldsymbol{x}_0 - \hat{\boldsymbol{x}}_0(\boldsymbol{x}_t, t, \boldsymbol{c})\|^2\right], \qquad (3)$$

where $\hat{\boldsymbol{x}}_0$ is the prediction of our world model and $\boldsymbol{c}$ is the

conditioning information which includes historical frames and the latent action $\tilde{a}$.

### 2.3. Highly Adaptable World Models

After action-aware pretraining across various environments, the world model can be controlled by different latent actions, making it highly adaptable for multiple applications, including efficient action transfer, world model adaptation, and even action creation.

**Efficient action transfer.** When presented with a demonstration video, we use the latent action encoder to extract a sequence of latent actions. This enables us to disentangle the action from its context and replicate it across different contexts. Specifically, given the initial frame from a new context, we can reuse the extracted latent action sequence as the conditions to generate a new video autoregressively. As demonstrated in Figure 4, AdaWorld naturally transfers actions from source videos to various contexts.

**Efficient world model adaptation.** AdaWorld also allows efficient world model adaptation with limited action labels and training steps. Specifically, after collecting a few action-video pairs through interactions, we use the latent action encoder to infer their latent actions. Thanks to the continuity of our latent action space, latent actions for the same label can be averaged directly. We empirically find that the averaged embedding consistently represents the intended action. Thus, for a new environment with $N$ discrete actions, we initialize a specialized world model using $N$ averaged latent actions and finetune the whole model for a few steps. For environments with continuous action spaces, since there are infinite options, we add a lightweight MLP to map raw action inputs to the latent action interface. The interface can also be efficiently initialized by finetuning the MLP with minimal action-latent action pairs. Figure 6 shows that the models initialized in aforementioned ways can be efficiently adapted to take control inputs through minimal finetuning.

**Action composition and creation.** It is also noteworthy

that AdaWorld enables several unique applications compared to existing world models. For instance, it allows the composition of new actions by interpolating observed actions within the latent space, as demonstrated in Figure 5. In addition, by collecting and clustering latent actions, we can easily create a flexible number of control options with distinct functions and strong controllability. This suggests that AdaWorld could serve as an alternative to generative interactive environments (Bruce et al., 2024). See Appendix C for experimental details on action creation.

# 3. Experiments

In this section, we first demonstrate AdaWorld's strengths in action transfer in Sec. 3.1. We then study how efficient world model adaptation enables better simulation and planning in Sec. 3.2. Lastly, we analyze the effectiveness of our designs with ablation studies in Sec. 3.3.

To thoroughly understand the adaptability of our approach, we compare AdaWorld with three representative baselines:

- **Action-agnostic pretraining.** In this setup, we train a world model that shares the same architecture as AdaWorld but always takes zeros as action conditions during pretraining. This baseline is used to demonstrate the effect of the predominant pretraining paradigm that relies only on action-agonistic videos (Mendonca et al., 2023; Wu et al., 2023; Gao et al., 2024; Che et al., 2025; Agarwal et al., 2025; He et al., 2025).
- **Optical flow as an action-aware condition.** We automatically predict optical flows from videos using Uni-Match (Xu et al., 2023a). The flow maps are downsampled to $16 \times 16$ and flattened as conditional encodings to replace the latent actions during pretraining. This baseline serves as an alternative solution for extracting action information from unlabeled videos.
- **Discrete latent action as an action-aware condition.** We also implement a variant of the latent action autoencoder based on the standard VQ-VAE (Van Den Oord et al., 2017). Instead of using a continuous latent action space, this variant adopts a VQ codebook with 8 discrete codes following Genie (Bruce et al., 2024).

Except for the above modifications, we align other training settings for the baselines and our method. The world models of all compared methods are trained for 50K iterations to ensure a fair comparison. Our training dataset comprises four publicly accessible datasets (Goyal et al., 2017; Grauman et al., 2022; O'Neill et al., 2024; Ju et al., 2024) and videos collected automatically from 1016 environments in Gym Retro (Nichol et al., 2018) and Procgen Benchmark (Cobbe et al., 2020). This results in about 2000 million frames of interactive scenarios in total. More details about our datasets and implementation are provided in Appendices A and B.

*Table 1.* **Action transfer comparison.** In both datasets, AdaWorld excels at transferring the demonstrated actions to different contexts.

| Method | LIBERO | | | SSv2 | | |
|---|---|---|---|---|---|---|
| | FVD↓ | ECS↑ | Human↑ | FVD↓ | ECS↑ | Human↑ |
| Act-agnostic | 1545.2 | 0.702 | 0% | 847.2 | 0.592 | 1% |
| Flow cond. | 1409.5 | 0.724 | 2% | 702.8 | 0.611 | 10.5% |
| Discrete cond. | 1504.5 | 0.700 | 3.5% | 726.8 | 0.596 | 21.5% |
| AdaWorld | 767.0 | 0.804 | 70.5% | 473.4 | 0.639 | 61.5% |

## 3.1. Action Transfer

AdaWorld can readily transfer a demonstrated action to various contexts without further training. Below, we provide both qualitative and quantitative evaluations to showcase how effectively AdaWorld performs action transfer.

**Qualitative results.** We transfer action sequences of length 20 through autoregressive generation in Figure 4. It shows that AdaWorld can effectively disentangle the demonstrated actions and emulate them across contexts. Qualitative comparison with other baselines can be found in Appendix C.

**Quantitative results.** To quantitatively compare with other baselines, we construct a evaluation set sourced from the unseen LIBERO (Liu et al., 2023) and Something-Something v2 (SSv2) (Goyal et al., 2017) datasets. Specifically, we select and pair videos from the same tasks in LIBERO and the same labels among the top-10 most frequent labels in SSv2, resulting in 1300 pairs for evaluation (more details in Appendix D). While the selected video pairs contain similar actions, we find that the video pairs from LIBERO often differ in the arrangement of objects, and those from SSv2 have significant differences in contexts. For each video pair, we take the first video as the demonstration video and use the first frame from the second video as the initial frame. We then generate videos by extracting action conditions from the demonstration video and employing different models to autoregressively predict the next 20 frames from the initial frame. The evaluation is performed by measure the generated videos against the original videos using Fréchet Video Distance (FVD) (Unterthiner et al., 2018). To complement the FVD evaluation, which reflects overall distribution similarity, we additionally employ Embedding Cosine Similarity (ECS) (Sun et al., 2024) that performs frame-level measurements with I3D (Carreira & Zisserman, 2017). We further conduct a human evaluation on a set of 50 video pairs from LIBERO and SSv2, respectively. Four volunteers are invited to judge whether the action is successfully transferred or not. Both the automatic and human evaluations in Table 1 demonstrate that our continuous latent action achieves the best action transfer performance, underscoring its capability to express more nuanced actions without losing generality.

## 3.2. World Model Adaptation

We also investigate how the proposed method benefits efficient world model adaptation in terms of simulation quality

*Table 2.* **Action-controlled simulation quality when adapting to four unseen environments.** All results are tested after 800 finetuning steps, using 100 samples for each discrete action (Habitat, Minecraft, DMLab) or 100 continuous trajectory samples (nuScenes).

| Method | Habitat (discrete action) | | Minecraft (discrete action) | | DMLab (discrete action) | | nuScenes (continuous action) | |
|---|---|---|---|---|---|---|---|---|
| | PSNR↑ | LPIPS↓ | PSNR↑ | LPIPS↓ | PSNR↑ | LPIPS↓ | PSNR↑ | LPIPS↓ |
| Act-agnostic | 20.34 | 0.450 | 19.44 | 0.532 | 20.96 | 0.386 | 20.86 | 0.475 |
| Flow cond. | 22.49 | 0.373 | 20.71 | 0.492 | 22.22 | 0.357 | 20.94 | 0.462 |
| Discrete cond. | 23.31 | 0.342 | 21.33 | 0.465 | 22.36 | 0.349 | 21.28 | 0.450 |
| AdaWorld | 23.58 | 0.327 | 21.59 | 0.457 | 22.92 | 0.335 | 21.60 | 0.436 |

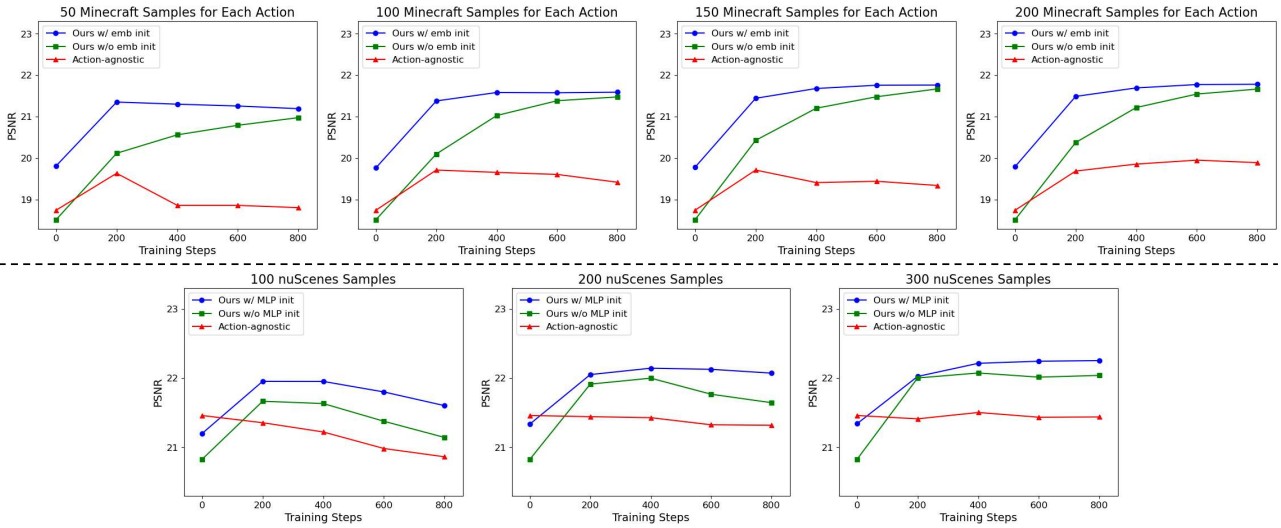

*Figure 6.* **PSNR curves for world model adaptation.** With limited samples and training steps, AdaWorld adapts to the action controls of new environments more rapidly than conventional pretraining methods.

and visual planning performance.

### 3.2.1. SIMULATION QUALITY

**Setup.** To evaluate the simulation quality after adaptation, we choose three environments with discrete action space (Habitat (Savva et al., 2019), Minecraft, DMLab (Beattie et al., 2016)) and one environment with continuous action space (nuScenes (Caesar et al., 2020)) that are not included in our training dataset. Each environment has a validation set consisting of 300 samples, which is used to evaluate the adaption quality in terms of PSNR (Hore & Ziou, 2010) and LPIPS (Zhang et al., 2018). To demonstrate the adaptability with restricted labels, we collect only 100 samples for each action in every discrete environment and 100 trajectories for nuScenes. Using the limited interaction data, we then finetune all compared world models for 800 steps with a batch size of 32 and a learning rate of $5 \times 10^{-5}$. The learning rate for the pretrained weights is discounted by a factor of 0.1. For the action-agnostic baseline, we initialize action embeddings with random parameters. For the other three models, we use the averaged action conditions extracted from the 100 samples to initialize the action embeddings as described in Sec. 2.3. Note that for nuScenes, we add a two-layer MLP to map continuous displacements to the latent action interface. The MLP is fine-tuned with limited

action-latent action pairs for 3K steps, which takes less than 30 seconds on a single GPU.

**Results.** As reported in Table 2, AdaWorld achieves the best fidelity after finetuning with limited interactions and compute. The comparison results suggest that the proposed method allows the world models to efficiently simulate new action controls in unseen environments. Note that all action-aware variants significantly outperform the action-agnostic baseline, underlining the importance of our key innovation, *i.e.*, incorporating action information during pretraining.

To further demonstrate our sample efficiency and finetuning efficiency, we conduct more comparative experiments using different sample numbers and finetuning steps on Minecraft and nuScenes. We show the evolving curves of PSNR in Figure 6. In all cases, AdaWorld performs much better at the beginning and improves significantly faster after a few finetuning steps. This suggests that our approach provides a superior initialization for efficient world model adaptation compared to conventional pretraining methods.

### 3.2.2. VISUAL PLANNING IN GAMES

**Setup.** After learning action controls, world models can be utilized for planning. To demonstrate the superiority of Ada-World in planning performance, we first compare it with the

*Table 3.* **Visual planning results in games.** For all selected scenes from four Procgen environments, we report the success rate and standard error with 5 random seeds. AdaWorld achieves higher average success rates than other baselines as well as Q-learning even without finetuning. Oracle : We use the ground truth simulator for MPC, which indicate the upper bound of this planning strategy.

| Method | Success Rate↑ | | | | |
|---|---|---|---|---|---|
| | Heist | Jumper | Maze | CaveFlyer | Average |
| Random | 19.33±4.41% | 22.00±2.50% | 41.33±5.44% | 22.00±2.50% | 26.17±2.55% |
| Act-agnostic | 20.67±3.55% | 20.67±2.45% | 39.33±2.87% | 23.33±1.84% | 26.00±0.98% |
| AdaWorld | | | | | |
|   w/o finetune | 38.67±2.01% | 68.00±2.25% | 41.33±2.72% | 31.33±2.50% | 44.83±1.37% |
|   w/ finetune | 66.67±4.09% | 58.67±2.50% | 68.00±1.69% | 33.33±3.80% | 56.67±2.16% |
| Q-learning | 22.67±3.87% | 47.33±6.71% | 4.67±0.81% | 34.00±6.17% | 27.17±1.27% |
| Oracle (GT env.) | 86.67±3.16% | 77.33±2.67% | 84.67±2.91% | 74.00±3.99% | 80.67±2.11% |

*Table 4.* **Visual planning results in robot tasks.** The success rates and standard errors are obtained over 4 runs for each task from $VP^2$. We also report the aggregated success rates normalized by the scores of the ground truth simulator on the right.

| Method | Success Rate↑ | | | | | | Aggregate |
|---|---|---|---|---|---|---|---|
| | Robosuite push | Open slide | Blue button | Green button | Red button | Upright block | |
| Act-agnostic | 17.50±0.50% | 1.67±1.67% | 5.00±1.67% | 3.33±0.00% | 0.00±0.00% | 1.67±1.67% | 5.03 |
| AdaWorld | 63.50±1.71% | 5.83±2.85% | 29.17±2.50% | 10.83±2.50% | 10.00±2.36% | 5.00±0.96% | 21.54 |

action-agnostic baseline in video game environments using sampling-based model predictive control (MPC) optimized by Cross-Entropy Method (De Boer et al., 2005; Chua et al., 2018). The MPC planning and optimization procedure are deferred to Appendix B.4. We define a goal-reaching task based on the Procgen benchmark (Cobbe et al., 2020) and select 30 scenes from each of four environments (Heist, Jumper, Maze, CaveFlyer). This ensures that the specified goals can be reached within an acceptable number of steps (more details in Appendix E). For each scene, we randomly collect 100 samples for each action in the default action space (LEFT, DOWN, UP, RIGHT). Based on the collected samples, the pretrained world models are finetuned for 500 steps with a batch size of 32 and a learning rate of $5 \times 10^{-5}$. We then use the finetuned world models to perform MPC planning in the selected scenes. The reward is defined as the cosine similarity between the predicted observations and the image of the final state. The planning is deemed successful if the agent reaches the final state within 20 steps.

**Results.** Table 3 presents the success rates averaged over 5 random seeds. While the action-agnostic baseline performs similarly to random planning, AdaWorld substantially increase the success rates across all environments. This indicates that our approach not only adapts more efficiently but also enables more effective planning. Visit our project page to see planning demonstrations of agents in games.

Additionally, we evaluate the visual planning performance without finetuning our model using the collected samples. In particular, we only utilize the averaged latent actions derived from these samples as action embeddings for the corresponding scenes. The results in Table 3 indicate that even without updating model weights, our variant still outperforms the finetuned action-agnostic pretraining baseline.

To further demonstrate the effectiveness of our approach, we also compare the planning results with Q-learning (Sutton & Barto, 2018), a classical model-free reinforcement learning method. For each scene, we construct a Q-table using the same samples collected for the MPC planning. The states of Q-table are represented by quantized images, and the rewards are obtained by computing the cosine similarity with the goal image. As shown in Table 3, AdaWorld significantly dominates the Q-learning method, suggesting that our approach makes more effective use of limited interactions.

### 3.2.3. VISUAL PLANNING IN ROBOT TASKS

**Setup.** To verify our efficacy in robot control tasks, we pretrain a low-resolution variant of AdaWorld and evaluate the planning performance on the $VP^2$ benchmark (Tian et al., 2023) after adaptation. The planning is also sampling-based and is performed using the model-predictive path integral (MPPI) (Williams et al., 2016; Nagabandi et al., 2020). We focus on a similar compute-efficient setting and finetune the pretrained variant and an action-agnostic baseline for 1K steps. The evaluation is conducted on 100 tabletop Robosuite tasks (Zhu et al., 2020) and 7 RoboDesk tasks (Kannan et al., 2021). More details are in Appendix B.5.

**Results.** Table 4 reports the success rates on $VP^2$. We omit the Flat block and Open drawer tasks from RoboDesk, as they do not yield meaningful scores under our constrained adaptation setting. The results show that AdaWorld adapts more efficiently with limited finetuning steps and improves the planning performance by a clear margin.

### 3.3. Ablation and Analysis

**Interface initialization.** We ablate the latent action initialization approaches in Sec. 2.3 with random initialization

*Table 5.* **Impacts of training data diversity.** Increasing data diversity enhances the generalization of latent actions to new domains.

| Training Data | Procgen | |
|---|---|---|
| | PSNR↑ | LPIPS↓ |
| OpenX | 25.51 | 0.318 |
| Retro | 26.43 | 0.250 |
| Retro+OpenX | **26.62** | **0.234** |

*Table 6.* **Generality of AdaWorld.** Applying action-aware pretraining to iVideoGPT also significantly improves its adaptability.

| Model | BAIR | |
|---|---|---|
| | PSNR↑ | LPIPS↓ |
| iVideoGPT | 16.59 | 0.220 |
| iVideoGPT+AdaWorld | **17.40** | **0.204** |

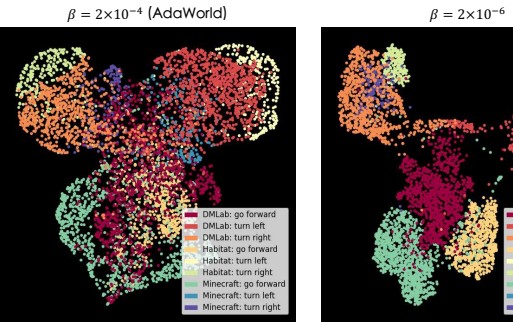

*Figure 7.* **UMAP of latent actions.** Reducing the value of $\beta$ increases expressiveness but sacrifices disentanglement from context.

and adapt the resultant model to the unseen Minecraft and nuScenes. Figure 6 shows that randomly initializing the control interface of AdaWorld results in a quality drop at the beginning. Nevertheless, it is noteworthy that although our variant is slightly worse than the action-agnostic pretraining baseline when the finetuning begins, it rapidly surpasses the action-agnostic baseline after just 200 steps. This is because AdaWorld has learned a highly adaptable control interface through action-aware pretraining, allowing it to efficiently adapt to an unseen environment by simply fitting the action embeddings for the new action space.

**Data diversity.** We also study the impact of data mixture to the generalization ability of latent actions. To this end, we implement three latent action autoencoder with different combinations of Open X-Embodiment (OpenX) (O'Neill et al., 2024) and Gym Retro (Nichol et al., 2018) datasets for 40K steps. We then assess the latent action decoder predictions of these three variants on Procgen benchmark (Cobbe et al., 2020) in Table 5. Surprisingly, we discover that even though OpenX mainly consists of real-world robot videos, incorporating OpenX helps the latent action autoencoder generalize to the unseen 2D virtual games in Procgen. This suggests that further increasing data diversity may positively affect the generalization of our latent actions.

**Method generality.** To demonstrate the generality of our method, we use iVideoGPT (Wu et al., 2024) as a state-of-the-art baseline. iVideoGPT is an action-controlled world model with an autoregressive Transformer architecture. It is pretrained by action-agnostic video prediction and adds a linear projection to learn action control during finetuning. For fair comparison, we implement a variant by conditioning iVideoGPT with our latent actions during pretraining. The training details can be found in Appendix B.6. After finetuning, we compare action-controlled simulation quality on BAIR robot pushing dataset (Ebert et al., 2017) in Table 6. The proposed action-aware pretraining significantly enhances the adaptability of iVideoGPT, suggesting that our method is generally applicable to different world models.

**Hyperparameter choice.** In Eq. (2), the hyperparameter $\beta$ is adjusted to achieve a good trade-off between expressiveness and context disentangling ability of latent actions. To provide a more intuitive illustration, we randomly collect 1000 samples for each action from Habitat, Minecraft, and DMLab, and use UMAP (McInnes et al., 2018) for visualization. Figure 7 shows that the same actions, even from different environments, are clustered together, which validates the context-invariant property of our latent actions. Note that noise exists because the action inputs cannot be executed in certain states (*e.g.*, cannot go ahead when an obstacle is in front). We also compare samples inferred by a model trained with a lower $\beta$. Although this results in more differentiable latent actions, it also reduces action overlap across environments thus sacrificing disentanglement ability. We therefore set $\beta$ as $2 \times 10^{-4}$ by default.

## 4. Conclusion

In this paper, we introduce *AdaWorld*, a new world model learning approach that facilitates efficient adaptation across various environments. It is highly adaptable in transferring and learning new actions with limited interactions and finetuning. Extensive experiments and analyses demonstrate the superior adaptability of AdaWorld, highlighting its potential as a new paradigm for world model pretraining.

**Limitations.** While AdaWorld promotes adaptable world modeling, several challenges remain. First, it does not operate at real-time frequency. Future work could incorporate distillation and sampling techniques (Feng et al., 2024; Yin et al., 2025) to accelerate inference speed. Similar to prior works (Yang et al., 2024e), AdaWorld struggles to create novel content when the rollout exceeds the initial scene. This issue is likely to be solved by scaling model and training data (Bruce et al., 2024; Bar et al., 2025). Additionally, our model falls short in achieving extremely long-term rollouts, and we will explore potential solutions (Chen et al., 2024a; Feng et al., 2024; Ruhe et al., 2024) in future work. We also append some primary failure cases in Appendix C.

## Acknowledgements

Shenyuan Gao and Jun Zhang were supported by the Hong Kong Research Grants Council under the NSFC/RGC Collaborative Research Scheme grant CRS_HKUST603/22.

## Impact Statement

This paper presents work whose goal is to advance the field of machine intelligence. There are many potential societal consequences of our work, none which we feel must be specifically highlighted here.

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

# A. Datasets

## A.1. Data Collection and Generation

Our training data is primarily sourced from four publicly accessible datasets (Goyal et al., 2017; Grauman et al., 2022; O'Neill et al., 2024; Ju et al., 2024) that represent various actions in typical interactive scenarios. For the videos from Open X-Embodiment (O'Neill et al., 2024), we keep both egocentric and exocentric demonstrations to encompass a wide spectrum of action patterns. Note that we manually remove the unfavorable subsets that are dominated by static, nonconsecutive, low-frequency and low-resolution frames.

To further enrich the diversity of our data, we also automatically generate a massive number of transitions from two gaming platforms developed by OpenAI (Nichol et al., 2018; Cobbe et al., 2020). We manually search on the Internet and import the ROMs we found into Gym Retro (Nichol et al., 2018), resulting in a total of 1000 interactive environments (see Table 8 for the full list). For the 16 environments in Procgen (Cobbe et al., 2020), we hold out 1000 start levels for evaluation and use the remaining 9000 levels for training. All samples are generated using the "hard" mode. We take a random action at each time step to collect 1M transitions for each Gym Retro environment and 10M for each Procgen environment. Unlike Genie (Bruce et al., 2024) that samples all actions uniformly in their case study, we employ a biased action sampling strategy to encourage broader exploration. Specifically, we increase the probability of selecting a particular action for a short period and then alternate these probabilities in the subsequent period. As shown in Figure 8, this simple strategy leads to much more diverse scenes in our data. Generating data with reinforcement learning agents may further boost the scene diversity (Kazemi et al., 2024; Yang et al., 2024d; Valevski et al., 2025), which we leave for future work. In Figure 9, we visualize some representative environments in our final dataset.

## A.2. Data Mixture

Since the datasets vary in size and diversity, it is challenging to balance them perfectly. Therefore, we simply weight all subsets according to the number of videos during training. We report detailed statistics of our training data in Table 7.

*Table 7.* **Data organization.** Data sources, generation procedures, approximated frame counts, and mixture ratios for our training.

| Category | Data Source | Automated | # Frames | Ratios |
|---|---|:---:|:---:|:---:|
| 2D Video Game | Gym Retro (Nichol et al., 2018) | ✓ | 1000M | 49% |
| | Procgen Benchmark (Cobbe et al., 2020) | ✓ | 144M | 2% |
| Robot Data | Open X-Embodiment (O'Neill et al., 2024) | ✗ | 170M | 30% |
| Human Activity | Ego4D (Grauman et al., 2022) | ✗ | 330M | 1% |
| | Something-Something V2 (Goyal et al., 2017) | ✗ | 7M | 3% |
| 3D Rendering | MiraData (Ju et al., 2024) | ✗ | 200M | 14% |
| City Walking | MiraData (Ju et al., 2024) | ✗ | 120M | 1% |

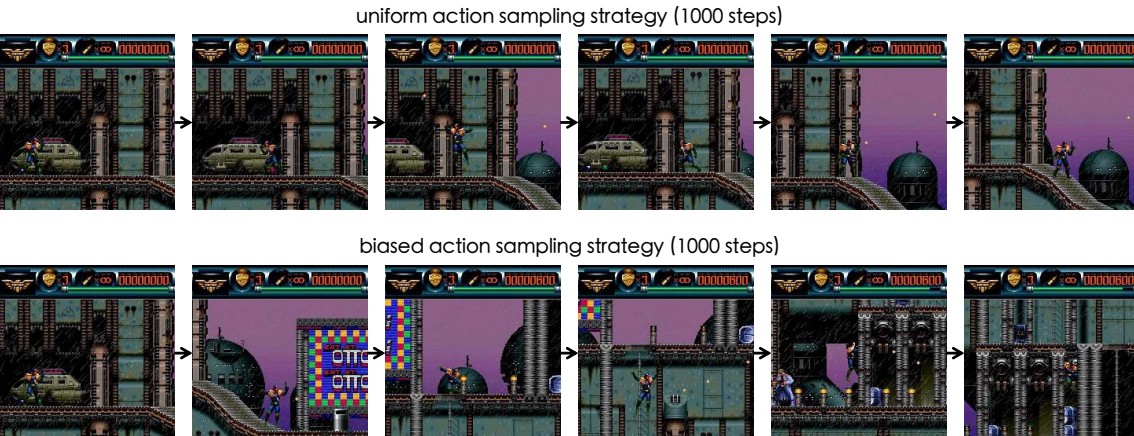

*Figure 8.* **Effect of our biased action sampling strategy.** Compared to the uniform action sampling strategy, our biased scheme enables agents to explore longer horizons.

*Table 8.* **Full list of collected Gym Retro environments.** We import 1000 ROMs from the web for automated data generation at scale.

## B. Implementation Details

### B.1. Architecture

The latent action autoencoder adopts a Transformer architecture with 500M parameters. It consists of 16 encoder blocks and 16 decoder blocks using 1024 channels and 16 attention heads. The dimension of the latent actions is 32. The autoregressive world model adopts a 3D UNet architecture following SVD (Blattmann et al., 2023), with 1.5B trainable parameters and a memory length of 6. The default input resolution for both models is $256 \times 256$.

### B.2. Training

The latent action autoencoder is trained for 200K steps from scratch with a batch size of 960. We employ the AdamW optimizer (Loshchilov & Hutter, 2019) with a learning rate of $2.5 \times 10^{-5}$ and a weight decay of 0.01. The hyperparameter $\beta$ is set to $2 \times 10^{-4}$ to achieve a good balance between representation capacity and context disentangling ability.

The autoregressive world model is trained for 80K steps with a batch size of 64 and a learning rate of $5 \times 10^{-5}$ on 16 NVIDIA A100 GPUs. We adopt a cosine learning rate scheduler with 10K warmup steps. To enhance prediction quality,

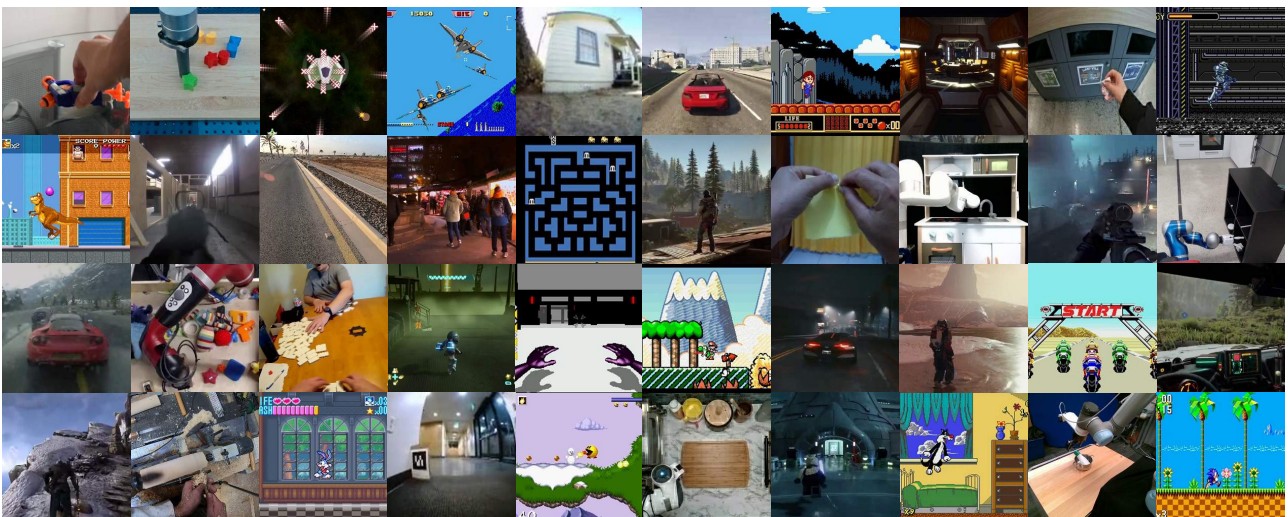

*Figure 9.* **Diversity of our training dataset.** Our curated dataset aggregates an extremely wide range of scenarios.

exponential moving average is also applied during pretraining. The noise augmentation level is randomly selected from a range of 0.0 to 0.7, with an interval of 0.1.

For both latent action autoencoder training and world model pretraining, we randomly jitter the brightness of input frames to augment generalization ability. For all frames with varying aspect ratios, we apply center cropping to avoid padding irrelevant content. We check the meta information of all datasets and downsample the videos to approximately a frame rate of 10 Hz for training.

### B.3. Sampling

By default, we use 5 sampling steps with a classifier-free guidance scale of 1.05 to generate new frames. We do not add noise to the historical frames but condition the world model on an augmentation level of 0.1 as we find this improves the results. A timestep shifting strategy (Kong et al., 2024) is also applied to enhance generation quality.

### B.4. Visual Planning on the Procgen Benchmark

We summarize our model predictive control process for visual planning as below:

1. Given the current observation and the image of the final state, where the environment is about to restart with a positive reward, we calculate the current reward as the cosine similarity between them in RGB space. The planning objective is defined as the maximum reward obtained along the planned trajectory.

2. At each iteration, we sample a population of $N$ action sequences, each with a length of $L$, from a distribution. The initial distribution is set to a uniform distribution over four actions (LEFT, DOWN, UP, RIGHT).

3. For each sampled action sequence, the world model is used to predict the resulting trajectory, and the reward is calculated for each trajectory.

4. The top $K$ action sequences with the highest rewards are selected, and we update the distribution by increasing the sampling probabilities of these selected actions.

5. A new set of $N$ action sequences is sampled from the updated distribution, and the process repeats for $i$ Cross-Entropy Method iterations.

6. After $i$ optimization iterations, the first $T$ action in the action sequence with the highest probability is executed in the environment. The planning terminates either when the final state is achieved or when the search limit is exceeded.

In practice, we use $i = 2$ Cross-Entropy Method iterations. For each iteration, $N = 100$ action sequences with a length of $L = 15$ are sampled, and the best $K = 10$ samples are selected to update the action sampling distribution. After the

optimization procedure is done, the first $T = 5$ actions are executed in the environment. We set the search limit to 20 steps. For efficiency, we use only 3 denoising steps and disable classifier-free guidance during planning.

### B.5. Visual Planning on the VP$^2$ Benchmark

We train low-resolution variants at a resolution of $64 \times 64$ for control-centric evaluation on the VP$^2$ benchmark. We follow the official protocol of VP$^2$ to evaluate all models. During adaptation, we use 5K given trajectories for Robosuite and 35K scripted trajectories with perturbations for RoboDesk for finetuning. Each world model is adapted for 1K steps with a batch size of 32 and a learning rate of $5 \times 10^{-4}$. A cost below 0.05 is considered as success in Robosuite tabletop pushing tasks.

### B.6. iVideoGPT Training Details

We resume from the official OpenX checkpoint of iVideoGPT and do not finetune its tokenizer. The model is designed to predict 15 future frames from an initial frame. After pretraining iVideoGPT and our action-aware variant on OpenX for 27K extra steps, we finetune each model with robot actions for 1K steps on BAIR robot pushing dataset. The resulting models are tested on 256 test videos.

## C. Additional Results

### C.1. Action Transfer

**Qualitative comparison.** We qualitatively compare AdaWorld trained for 50K steps with other baselines in Figure 10. The results show that our approach is able to represent nuanced actions and exhibit strong transferability across contexts.

**Visualizations.** We attach more action transfer results in a number of environments in Figures 12 to 16. Each sample is generated by transferring a latent action sequence with a length of 20.

**Failure case study.** We present typical failure cases of action transfer in Figure 21. AdaWorld does not always perfectly understand physics and dynamics. It also struggles to produce high-quality content during long-term rollouts or dramatic view shifts.

### C.2. World Model Adaptation

To demonstrate the advantages of our approach, in Figure 11, we visualize some simulation results on the held-in test set using the finetuned models in Sec. 3.2.1. While the model pretrained with action-agnostic videos struggles to faithfully execute the action inputs, our model has rapidly acquired precise action controllability using the same number of steps. This underscores that our approach could serve as a better initialization method for world model adaptation compared to conventional methods.

### C.3. Action Creation through Clustering

As mentioned in Sec. 2.3, AdaWorld can also easily create a flexible number of control options through latent action clustering. Specifically, we process our Procgen and Gym Retro training set using the latent action encoder to obtain the corresponding latent actions. To generate different control options, we apply K-means clustering to all latent actions, setting the number of clustering centers to the desired number of control options. To examine the controllability of varying actions derived with AdaWorld, we adopt the $\Delta$PSNR metric following Genie (Bruce et al., 2024). Table 9 shows the $\Delta$PSNR of the latent action decoder predictions. The larger the $\Delta$PSNR, the more the predictions are affected by the action conditions and therefore the world model is more controllable. The results in Table 9 demonstrate that the control options derived with AdaWorld represent distinct meanings and exhibit comparable controllability to the discrete counterpart, while the latter does not support a customizable number of actions, as it is fixed once trained.

*Table 9.* **Action controllability evaluation.** AdaWorld supports customizing different numbers of actions with strong controllability.

| Method | $\Delta$PSNR | | | | | | | | |
|---|---|---|---|---|---|---|---|---|---|
| | 4 | 5 | 6 | 7 | 8 | 9 | 10 | 11 | 12 |
| Discrete cond. | N/A | N/A | N/A | N/A | 6.47 | N/A | N/A | N/A | N/A |
| AdaWorld | 5.67 | 5.15 | 7.28 | 8.23 | 6.26 | 7.32 | 6.07 | 6.68 | 6.53 |

# D. Something-Something v2 Categories for Action Transfer

In Sec. 3.1, we utilize the top-10 most frequently appearing categories from SSv2 for action transfer evaluation, including "Putting [something] on a surface", "Moving [something] up", "Pushing [something] from left to right", "Moving [something] down", "Covering [something] with [something]", "Pushing [something] from right to left", "Uncovering [something]", "Taking [one of many similar things on the table]", "Throwing [something]", and "Putting [something] into [something]".

# E. Selected Scenes for Visual Planning

To guarantee an effective evaluation, we conduct an exhaustive search using the ground truth environments to identify viable Procgen test scenes that can be completed in an acceptable number of steps by goal image matching. This results in a total of 120 scenes for our visual planning experiments in Sec. 3.2.2. In Figures 17 to 20, we illustrate the initial states of all selected scenes. The scenes from survival-oriented environments (*e.g.*, BigFish and BossFight) are not considered in our evaluation.

# F. Related Work

## F.1. World Models

Driven by advances in deep generative models (Peebles & Xie, 2023; Yang et al., 2025), world models have made encouraging strides recently. They aim to replicate the transitions of the world through generation (Sutton, 1991; Ha & Schmidhuber, 2018; Yang et al., 2024c; Lu et al., 2024; 2025; Agarwal et al., 2025), enabling agents to perform decision making entirely in imagination even when confronted with unseen situations (Kaiser et al., 2020; Micheli et al., 2023; Hafner et al., 2023; Du et al., 2023; Mazzaglia et al., 2024). This capability has proven beneficial for gaming agents (Kim et al., 2020; Alonso et al., 2024; Pearce et al., 2024; He et al., 2025), autonomous vehicles (Kim et al., 2021; Hu et al., 2023; Yang et al., 2024a; Gao et al., 2024; Wang et al., 2024; Bar et al., 2025; Hassan et al., 2025), and real robots (Seo et al., 2023; Wu et al., 2022; Ko et al., 2024; Du et al., 2024; Zhen et al., 2024; Zhou et al., 2024b; Wu et al., 2024; Zhu et al., 2024; Zhou et al., 2024a; Bu et al., 2024; Wang et al., 2025; Qi et al., 2025; Wu et al., 2025).

Despite the substantial benefits, existing works often suffer from costly training that requires extensive annotations to learn new actions in different environments. To mitigate this issue, some efforts have explored more efficient world modeling through pretraining from passive videos (Watter et al., 2015; Seo et al., 2022; Mendonca et al., 2023; Wu et al., 2023). However, these methods primarily focus on obtaining compact world representations rather than learning control interfaces that can be readily adapted with limited interactions and computations. Another family of research investigates efficient online adaptation (Yang et al., 2024b; Rigter et al., 2024; Hong et al., 2025), but they primarily focus on parameter efficiency by assuming that the parameters of the pretrained video models are frozen or inaccessible.

While a recent study (Bruce et al., 2024) has explored learning interactive environments from 2D gameplay videos, it is limited to 8 fixed actions, which constrains its ability to express and adapt to more fine-grained actions. In contrast, we advocate for using a continuous latent action space, maximizing flexibility for efficient action transfer and enabling several unique applications for adaptable world modeling.

## F.2. Latent Action from Videos

Humans can comprehend the notion of various actions by observing others to behave (Rizzolatti et al., 1996; Rybkin et al., 2019; Schmeckpeper et al., 2020; Yatim et al., 2024; Ling et al., 2025). To develop intelligent agents, it is intriguing to automatically extract similar action primitives from videos. Some approaches align human and robot manipulation videos in a shared space to derive meaningful action prototypes (Xu et al., 2023b). However, they require collecting semantically paired videos, which limits their scalability for most real-world tasks.

To avoid reliance on video pairs, some works extract implicit latent actions through fully unsupervised learning (Edwards et al., 2019; Menapace et al., 2021; 2022; Ye et al., 2023; Schmidt & Jiang, 2024; Zhang et al., 2024; Sun et al., 2024; Villar-Corrales & Behnke, 2025). However, they mainly focus on a single environment with limited complexity. While more recent works extend similar approaches to multiple datasets (Cui et al., 2024; Chen et al., 2024b;c; Ye et al., 2025; Bu et al., 2025; Ren et al., 2025), they mainly leverage latent actions as the objectives for behavior cloning. Consequently, they always adopt discrete actions to fit the policy networks (Kim et al., 2024), leading to significant ambiguity due to discretization (Nikulin et al., 2025). Thus, the potential of latent actions for adaptable world modeling remains underexplored.

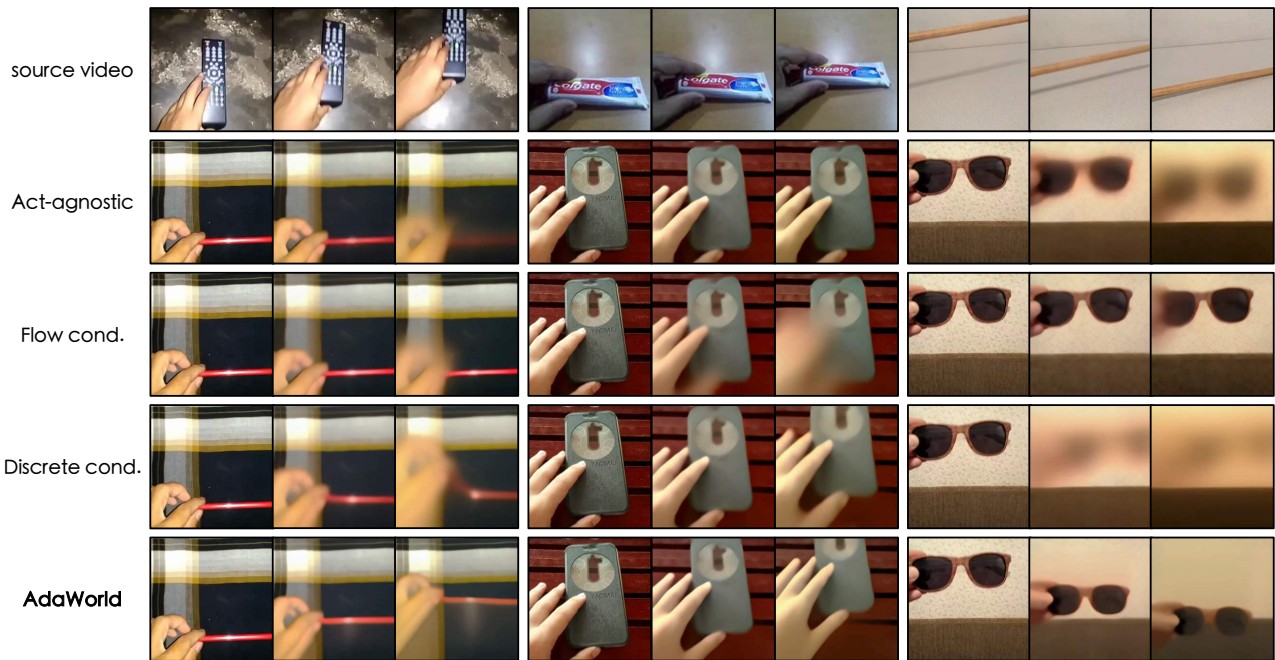

*Figure 10.* **Qualitative comparison of action transferability.** AdaWorld can accurately identify the demonstrated action and precisely transfer it to another context, while the other baselines fall short in doing so.

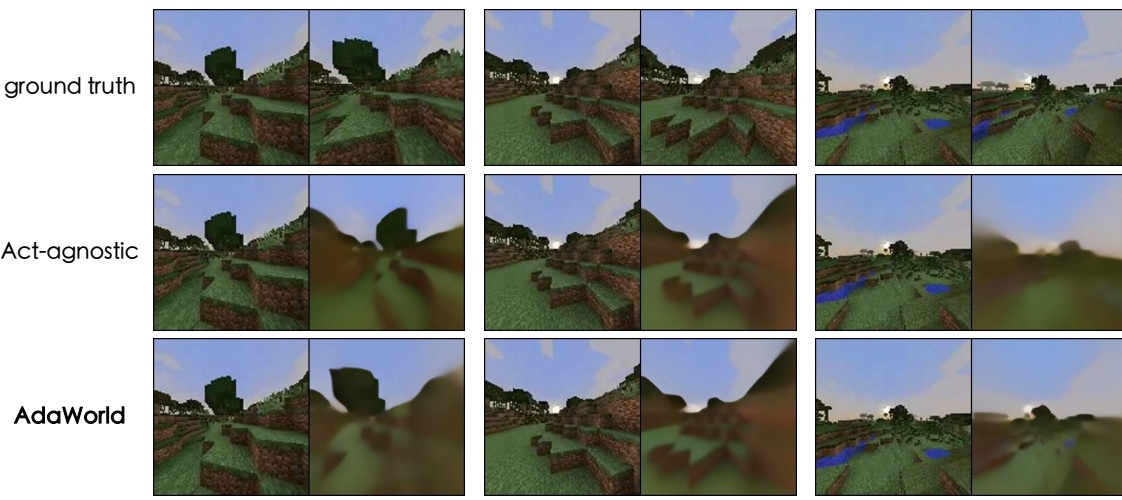

*Figure 11.* **Minecraft adaptation results after finetuning 800 steps with 100 samples for each action.** Our approach efficiently achieves precise action controllability with minimal action-labeled data and finetuning, while the action-agnostic pretraining baseline fails to perform the correct actions.

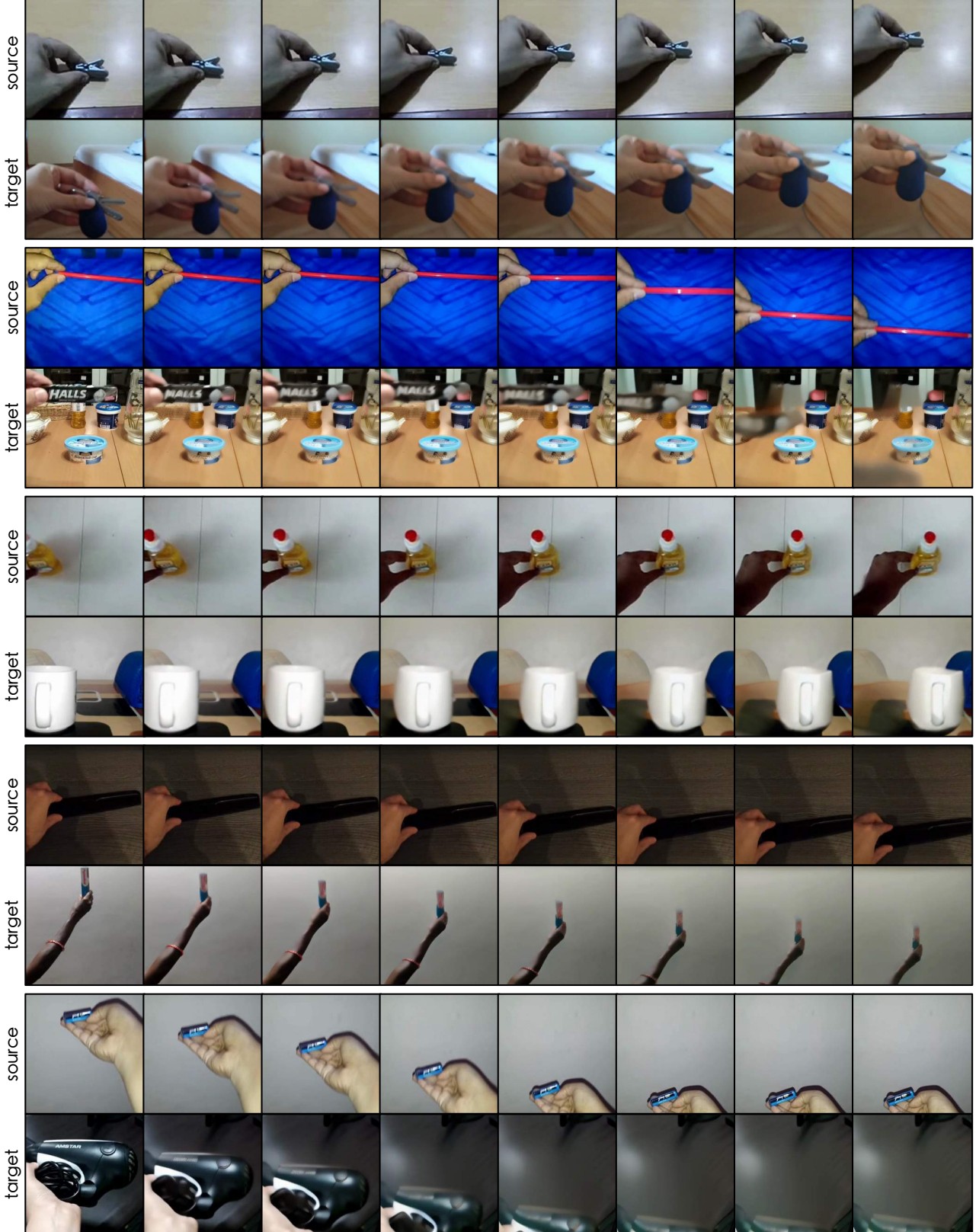

*Figure 12.* **Additional action transfer results by AdaWorld.** Best viewed with zoom-in.

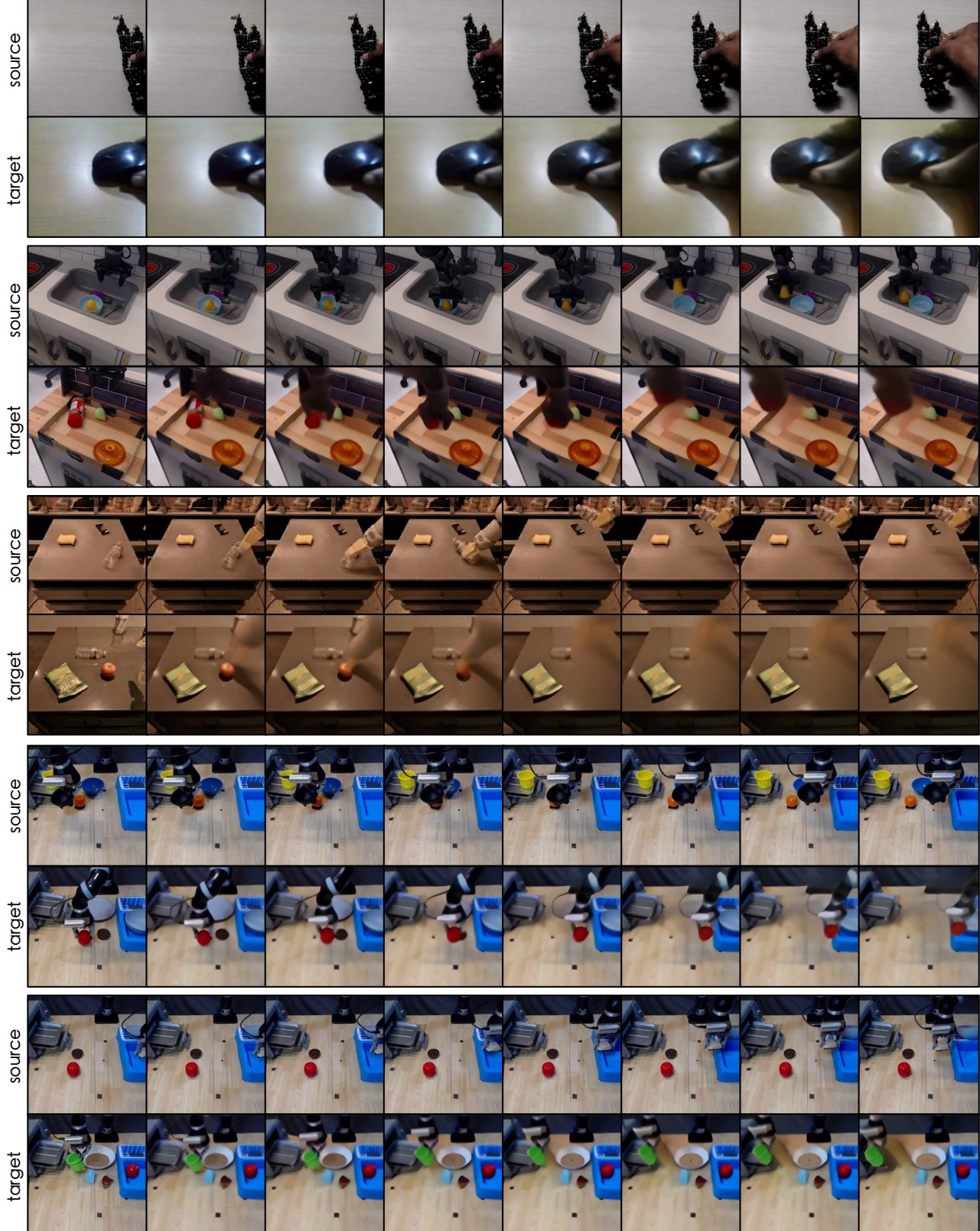

*Figure 13.* **Additional action transfer results by AdaWorld.** Best viewed with zoom-in.

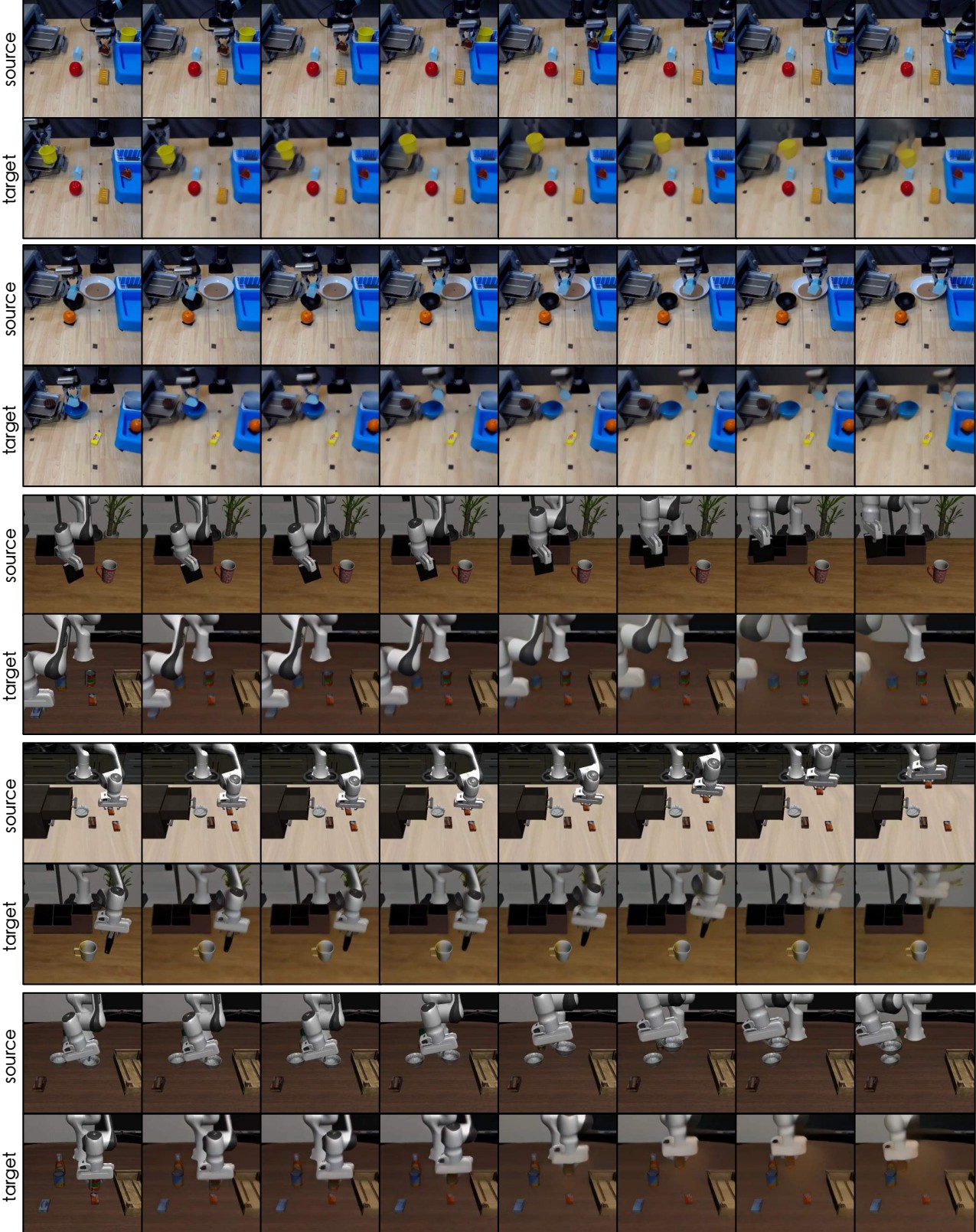

*Figure 14.* **Additional action transfer results by AdaWorld.** Best viewed with zoom-in.

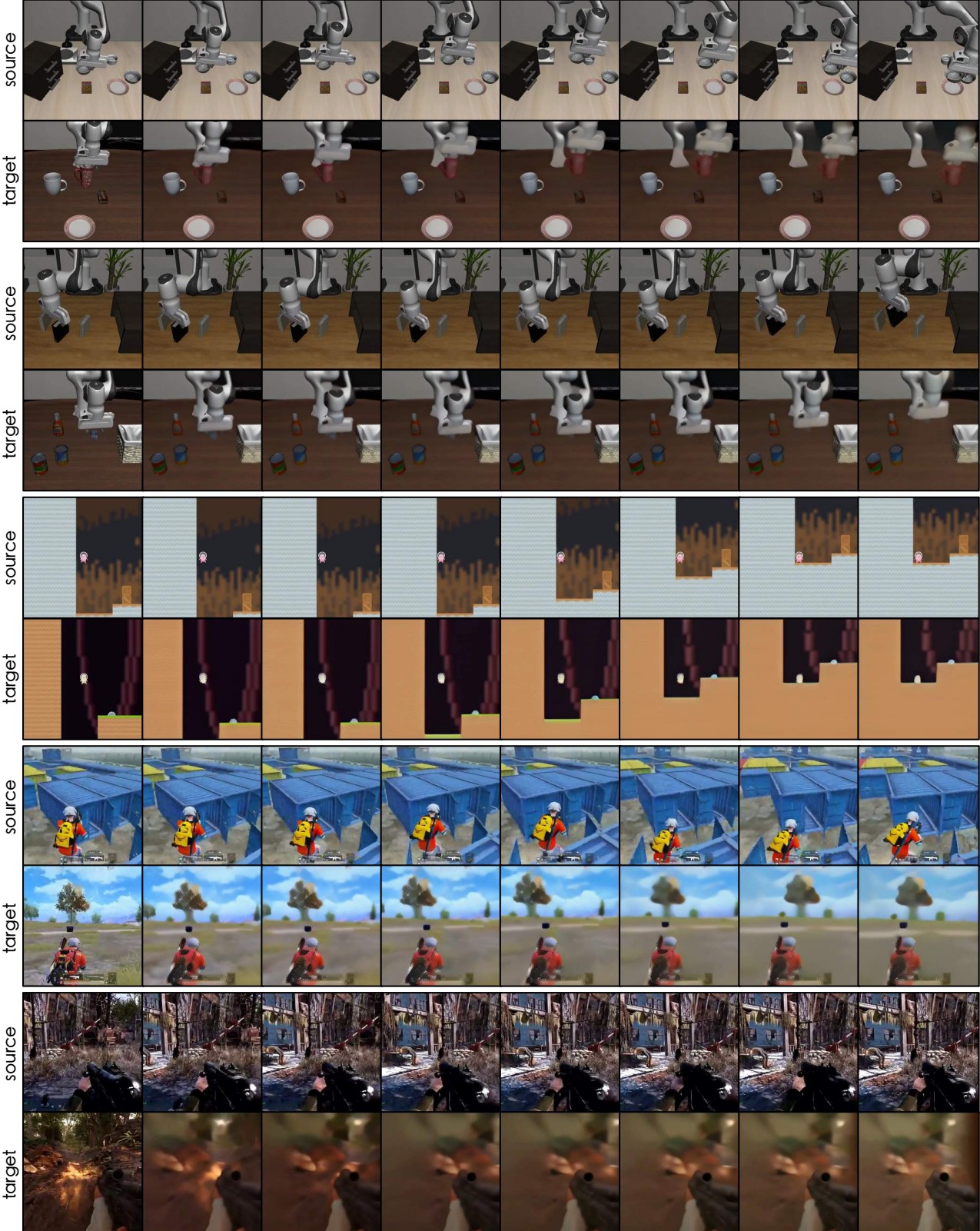

*Figure 15.* **Additional action transfer results by AdaWorld.** Best viewed with zoom-in.

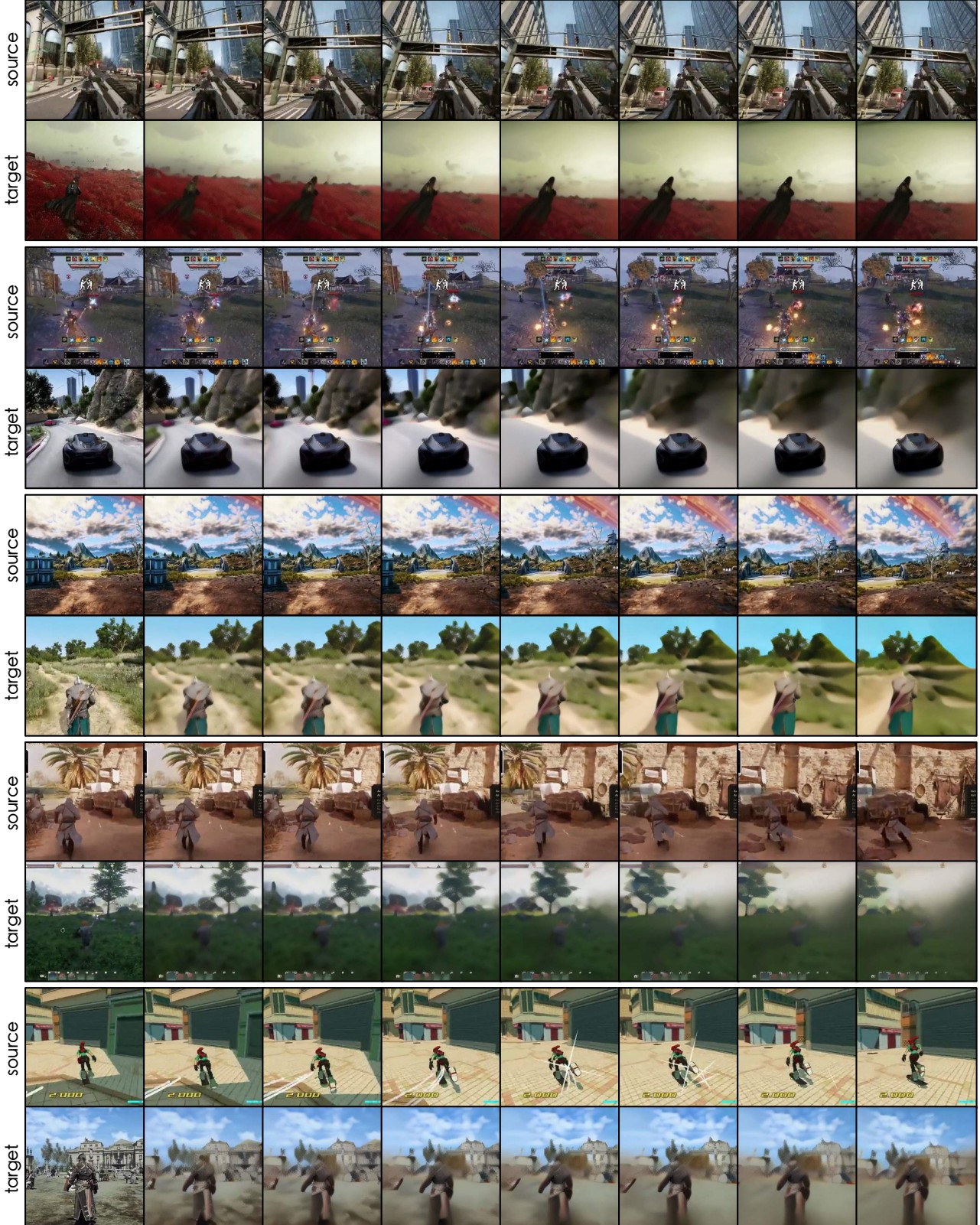

Figure 16. **Additional action transfer results by AdaWorld.** Best viewed with zoom-in.

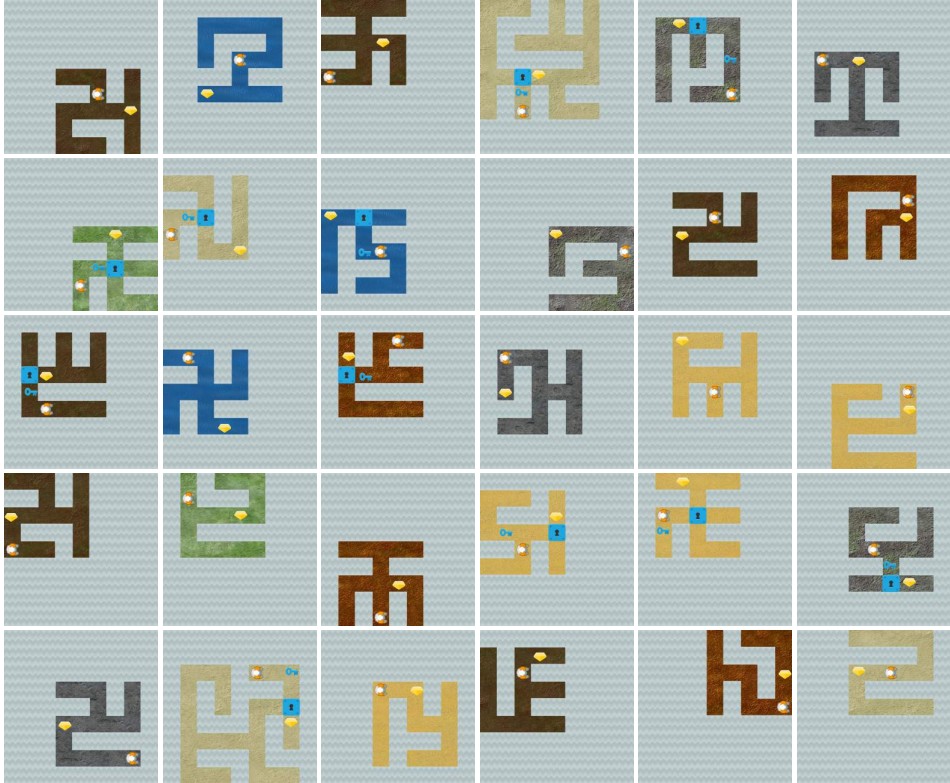

Figure 17. **Heist scenes for planning evaluation.** The figure illustrates the initial states of all selected scenes.

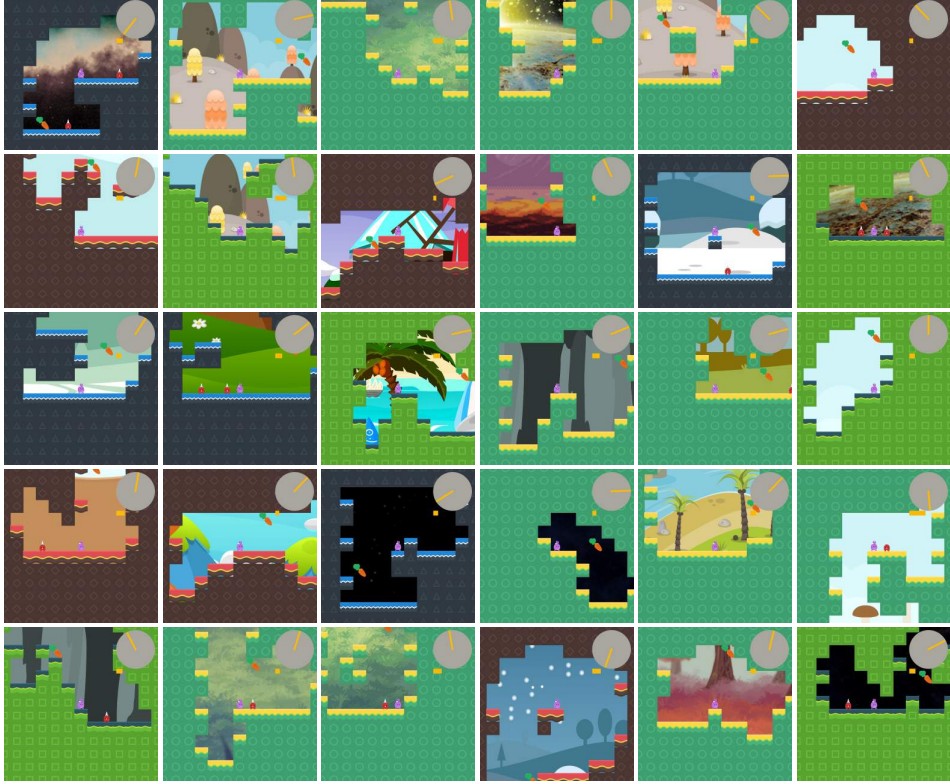

Figure 18. **Jumper scenes for planning evaluation.** The figure illustrates the initial states of all selected scenes.

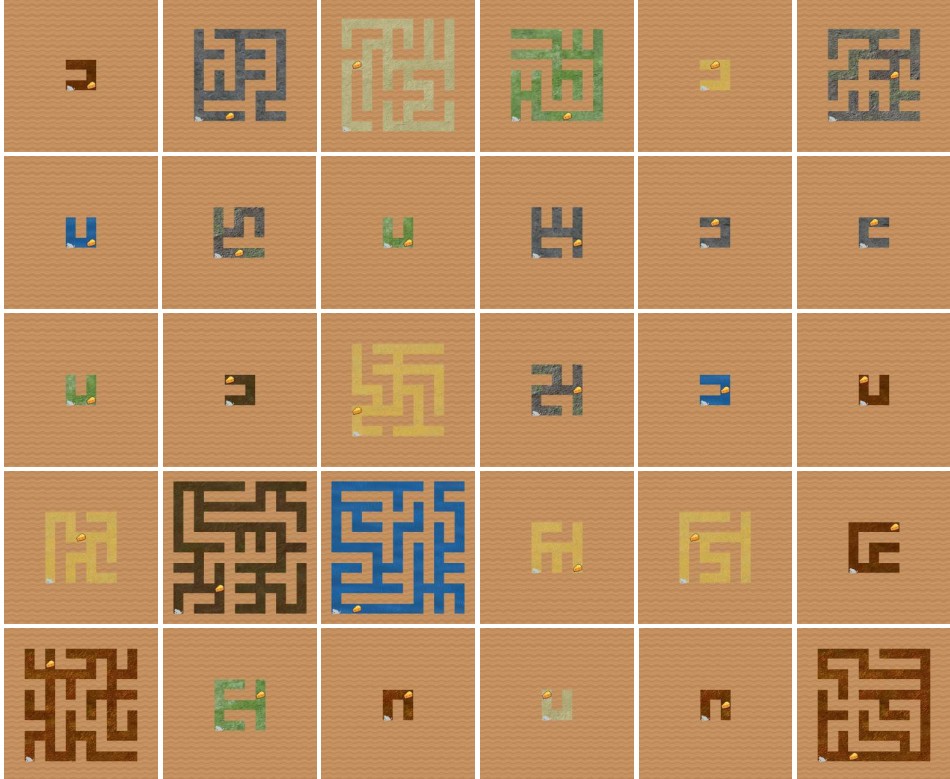

Figure 19. **Maze scenes for planning evaluation.** The figure illustrates the initial states of all selected scenes.

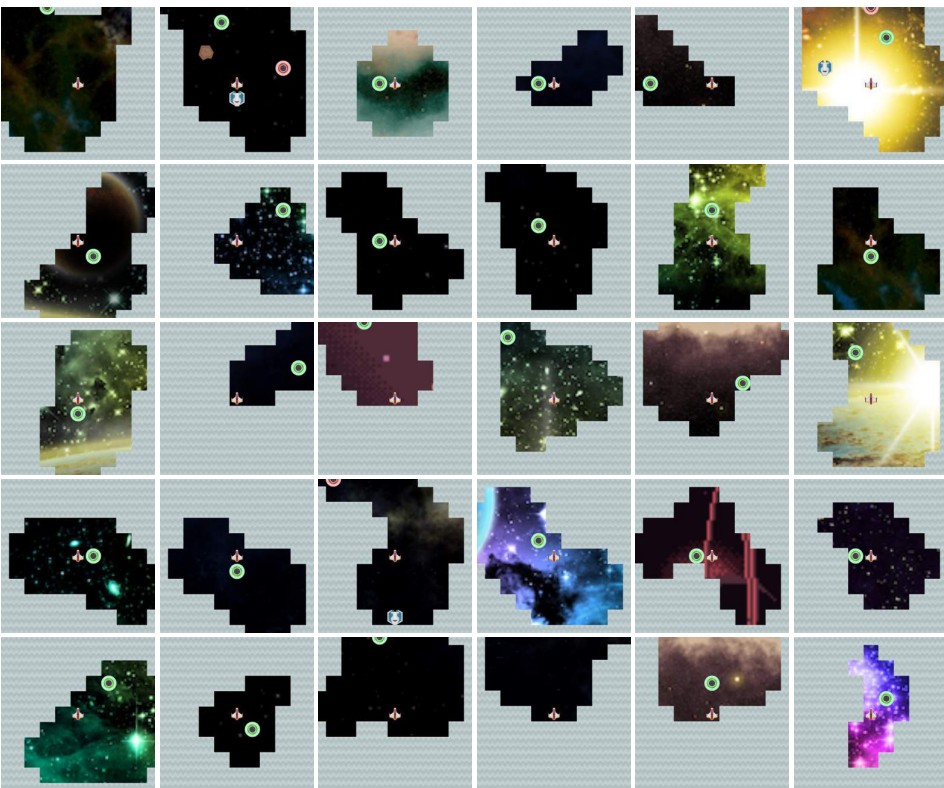

Figure 20. **CaveFlyer scenes for planning evaluation.** The figure illustrates the initial states of all selected scenes.

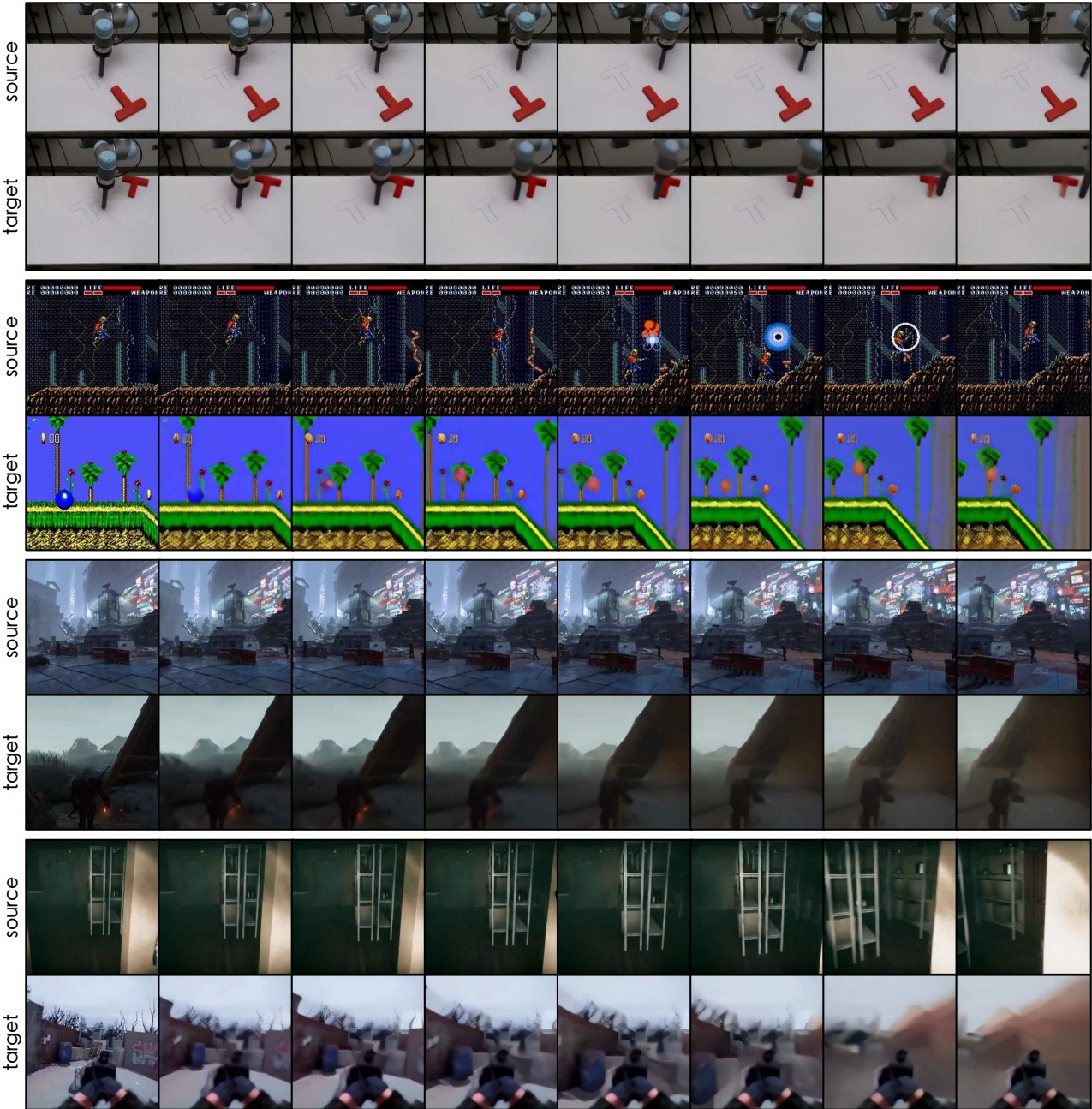

*Figure 21.* **Examples of failure cases.** AdaWorld is by no means perfect in simulating real-world physics, dynamic agents, long-term rollouts, and significant view changes.

