# OpenReview forum: "AdaWorld: Learning Adaptable World Models with Latent Actions"
_ICML.cc/2025/Conference — ICML 2025 poster_

### Official Review · Reviewer_jpmM · 2025-03-06

**Overall Recommendation:** 3

**Summary:**

This paper focuses on learning world models from general videos. Unlike previous approaches that solely rely on video-based learning, this work extracts latent actions in a self-supervised manner and leverages action information for large-scale world model pretraining. With the aid of latent actions, the model can efficiently transfer actions across different contexts and adapt to new environments with limited interactions. Comprehensive experiments are conducted across multiple environments to validate the effectiveness of AdaWorld.

**Claims And Evidence:**

This work claims that by learning latent actions:

1. AdaWorld can directly transfer actions to different contexts.
2. It can efficiently adapt into specialized world models with limited interactions and fine-tuning.

The authors validate these claims through experiments and visualizations. Figure 2 demonstrates action transfer to support Claim 1, while Table 1&2 showcases the model's rapid adaptation capabilities. Compared to other conditioning methods, AdaWorld achieves superior performance.

**Essential References Not Discussed:**

NA

**Experimental Designs Or Analyses:**

The paper evaluates the approach through both quantitative and qualitative experiments. Tables 1 & 2 validate the advantages of prediction under the action latent space. Overall, the experimental setup is reasonable. However, it lacks visualization or feature similarity analysis of the learned action latent representations.

**Methods And Evaluation Criteria:**

The method employs a β-VAE to learn latent actions and a conditional diffusion model for video generation. The evaluation primarily focuses on the consistency and quality of generated videos, with FVD and ECS serving as reasonable metrics.

**Other Comments Or Suggestions:**

1. The paper validates the benefits of the learned world model through visual planning. A more convincing approach could be testing in robotics simulators, where a goal image is provided for generation, followed by evaluating the corresponding planning performance using a video-action model.
2. Providing an analysis of the action latents would make the findings more convincing.

**Other Strengths And Weaknesses:**

1. The learned latent actions seem quite similar to those in Genie and Genie2, primarily representing simple movements such as up, down, left, and right. However, they struggle to effectively capture complex and long-range motion representations.

**Questions For Authors:**

1. From the visualizations, a decline in generation quality over time can be observed. How does the model perform when generating longer temporal sequences?

**Relation To Broader Scientific Literature:**

This paper follows a similar idea to Genie, aiming to learn action latents in a self-supervised manner to improve world model learning. Additionally, the approach of learning difference information between two given frames is commonly used in the robotics field.

**Theoretical Claims:**

No theoretical claims.

---

> ### Author Rebuttal · Authors · 2025-04-01
>
> Thanks for the constructive feedback. We answer each question below and will include all results and discussions in the revision.
>
> > A visualization or feature similarity analysis of the action latents.
>
> **R4-1**: As suggested, we randomly collect 1000 samples for each action from three environments (Habitat, Minecraft, DMLab) and use UMAP projection [1] to visualize them [[LINK](https://icml2025-1014.github.io)]. The visualization shows that the same actions, even from different environments, are clustered together, which validates the context-invariant property of our latent actions. Note that noise exists because the action inputs cannot be executed in certain states (e.g., cannot go ahead when an obstacle is in front). We also compare the latent action autoencoder trained with a different hyperparameter choice in the right figure. Although a lower $\beta$ results in more differentiable latent actions, it also reduces action overlap across environments thus sacrificing disentanglement ability.
>
> > This paper follows a similar idea to Genie, aiming to learn action latents in a self-supervised manner to improve world model learning. Additionally, the approach of learning difference information between two given frames is commonly used in the robotics field.
>
> **R4-2**: While the concept of latent action is not new, we are the first to find its usefulness in world model pretraining. Unlike existing works that mainly adopt a discrete action set for imitation learning or playability, we propose a continuous latent action space that enables more effective adaptation. As mentioned in Sec. 2.3, our design also enables several unique applications compared to prior works like Genie, such as action composition and clustering.
>
> > The learned latent actions seem quite similar to those in Genie and Genie2, primarily representing simple movements such as up, down, left, and right. However, they struggle to effectively capture complex and long-range motion representations.
>
> **R4-3**: Unlike Genie, which is limited to 8 fixed actions, we develop a continuous latent action space that can express a wide range of diverse actions. Compared to Genie’s discrete design, our model can capture and transfer more nuanced actions (Table 1). Since our model is pretrained with various kinds of actions, adapting it to a new environment is akin to matching the corresponding latent actions for the action space, which leads to better simulation quality (Table 2). We also add some action transfer results showing that our model captures complex actions [[LINK](https://icml2025-1014.github.io)].
>
> Regarding the range of actions, our current model opts for frame-level control to achieve finer granularity. In future work, we will explore extending our general training recipe to long-range settings, e.g., predicting multiple frames with one latent action.
>
> > Visual planning evaluation in robotics simulators, where a goal image is provided for generation, followed by evaluating the corresponding planning performance using a video-action model.
>
> **R4-4**: Thanks for the suggestion. We use the 100 robosuite control tasks from the VP2 benchmark [2] to evaluate the effectiveness of our method in robotic tasks. We focus on a compute-efficient setting and finetune our pretrained world model and the action-agnostic baseline for only 1k steps. The finetuned models are then used to perform goal-conditioned model predictive control following the official protocol of the VP2 benchmark. The success rates are reported below:
>
> | | robosuite |
> | :--- | :---: |
> | Act-agnostic | 14% |
> | AdaWorld | **61%** |
>
> The results demonstrate that our action-aware pretraining significantly enhances robot planning performance with limited finetuning steps. We will experiment with more robotic environments in the revision.
>
> > How does the model perform when generating longer temporal sequences?
>
> **R4-5**: The current model can be stably controlled for about 20 frames. Generating longer sequences is likely to result in quality degradation. While this paper mainly focuses on enabling adaptable world models, we believe our study complements other progresses in this field. We will explore potential solutions for long-horizon rollouts in future work.
>
> [1] UMAP: Uniform Manifold Approximation and Projection for Dimension Reduction
>
> [2] A Control-Centric Benchmark for Video Prediction

---

### Official Review · Reviewer_vV4B · 2025-03-14

**Overall Recommendation:** 3

**Summary:**

The authors propose a method to incorporate latent actions into the pre-training stage of World-models allowing for more efficient adaptation to downstream tasks. The authors curate a dataset spanning from ego perspectives and third-person views to virtual games and real-world environments. Finally, they evaluate their model on action transfer, model adaptation and visual planning.

**Claims And Evidence:**

Yes the authors' claims are evidence backed. The action aware pre-training does show promising results in action transfer, model adaptation and visual planning.

**Essential References Not Discussed:**

NA

**Experimental Designs Or Analyses:**

In line 317-319 authors say that "We randomly initialize action embeddings for the action-agnostic
video pretraining baseline." However in lines 282-285 the authors say "pretrain a world model that shares the same architecture as AdaWorld but does not take latent actions as conditions." These two statements seem inconsistent.

**Methods And Evaluation Criteria:**

The authors mention using two action tokens  $a_{t:t+1}$  during the latent action autoencoding but they mention that they use $a_{t+1}$ to approx the posterior. What do the authors do with $a_{t}$ ? Also the authors mention that in the action aware pretraining, the next frame is predicted based on a sequence of the latent actions. Which latent action $a_{t+1}$ or $a_{t}$? These details need to be carefully described and mathematically formulated instead of just via text.

**Other Comments Or Suggestions:**

NA

**Other Strengths And Weaknesses:**

NA

**Questions For Authors:**

see above

**Relation To Broader Scientific Literature:**

The proposed action aware pre-training is quite novel although encoding latent actions from videos is not.

**Theoretical Claims:**

NA

---

> ### Author Rebuttal · Authors · 2025-04-01
>
> Thanks for the helpful feedback. We answer each question below and will include all results in the revision.
>
> > The authors mention using two action tokens $a_{t:t+1}$ during the latent action autoencoding but they mention that they use $a_{t+1}$ to approximate the posterior. What do the authors do with $a_{t}$? Also the authors mention that in the action-aware pretraining, the next frame is predicted based on a sequence of the latent actions. Which latent action $a_{t+1}$ or $a_{t}$? These details need to be carefully described and mathematically formulated instead of just via text.
>
> **R3-1**: Sorry for the confusion. The $a_{t}$ ensures that the total number of tokens is aligned between $t$ and $t+1$, simplifying the implementation of spatiotemporal attention. Let $f_{t}$ and $f_{t+1}$ represent image tokens and [;] denote concatenation operation. The spatial attention can be formulated as:
>
> [$f_{t}$’;$a_{t}$’] = SpatialAttn([$f_{t}$;$a_{t}$]),
>
> [$f_{t+1}$’;$a_{t+1}$’] = SpatialAttn([$f_{t+1}$;$a_{t+1}$]),
>
> and the temporal attention can be formulated as:
>
> [$f_{t}$’;$f_{t+1}$’] = TemporalAttn([$f_{t}$;$f_{t+1}$]),
>
> [$a_{t}$’;$a_{t+1}$’] = TemporalAttn([$a_{t}$;$a_{t+1}$]).
>
> After encoding, all tokens except for the $a_{t+1}$ will be discarded, and we only project $a_{t+1}$ to estimate the posterior. Formally,
>
> [$\mu$;$\sigma$] = FC($a_{t+1}$).
>
> Thus, for two consecutive frames $f_{t:t+1}$ in the video sequence, the corresponding latent action is predicted from $a_{t+1}$. We will revise this part accordingly and open-source all code for reproducibility.
>
> > In line 317-319 authors say that "We randomly initialize action embeddings for the action-agnostic video pretraining baseline." However in lines 282-285 the authors say "pretrain a world model that shares the same architecture as AdaWorld but does not take latent actions as conditions." These two statements seem inconsistent.
>
> **R3-2**: Our latent action is concatenated with the timestep embedding and CLIP image embedding in the original SVD, which is equivalent to adding a linear projection layer to these two layers. For the action-agnostic baseline, we use the same architecture but input zeros into the additional linear projection during pretraining. When adapting to the discrete action space, the inputs are chosen from an action codebook. We utilize the averaged latent actions to initialize the embeddings of this codebook. Since the latent actions are not learned in the action-agnostic baseline, we initialize its action codebook with random parameters. We will clarify this in the revision.

---

> > ### Comment · Reviewer_vV4B · 2025-04-04
> >
> > I thank the authors for clarifying their approach. I will maintain my previous score.

---

### Official Review · Reviewer_UZ2f · 2025-03-15

**Overall Recommendation:** 3

**Summary:**

This paper proposes a pretraining framework for learning world models that can generalize to various contexts.  AdaWorld first learns a latent action representation using an unsupervised forward prediction objective. Subsequently, AdaWorld learns an autoregressive world model that conditions on latent actions to produce future frames. Interestingly, AdaWorld finds that the learned latent actions are context-invariant, allowing for transfer between contexts. AdaWorld is evaluated on a variety of downstream tasks in action transfer, model adaptation, and visual planning.

**Claims And Evidence:**

Yes, the paper claims that the learned world model can generalize to new, unseen contexts. The experiments show that by applying latent actions from one demonstrations, AdaWorld can generalize the same behavior to generate behavior for a unseen scene conditioned on just a single frame. There are also experiments demonstrating AdaWorld's capabilities on visual planning and adaption. However, these experiments are not as comprehensive and could use additional comparisons to state-of-the-art baselines.

**Essential References Not Discussed:**

[1] Ye, Seonghyeon, et al. "Latent action pretraining from videos." arXiv preprint arXiv:2410.11758 (2024).
[2] Menapace, Willi, et al. "Playable video generation." Proceedings of the IEEE/CVF Conference on Computer Vision and Pattern Recognition. 2021.

**Experimental Designs Or Analyses:**

To evaluate their approach, they measured AdaWorld's ability to generalize to new contexts on a single demonstration. The paired data between LIBERO and Something Something v2 is reasonable. Also the evaluation environments used are quite common in the literature.

**Methods And Evaluation Criteria:**

Yes, the proposed method is sound and the evaluation is based on standard metrics used in the literature.

**Other Comments Or Suggestions:**

See weaknesses and questions section.

**Other Strengths And Weaknesses:**

Strengths:
- Paper is well written and easy to follow
- Implementation details and framework is described thoroughly
- AdaWorld has diverse set of applications in task transfer and visual planning

Weakness:
- Method is technically not very novel. Prior work has already proposed latent action models in LAPO and Genie and similarly there is a plethora of work on generative video models
- No state-of-the-art baselines. Baseline methods are mostly ablations or modifications to the approach (e.g. predicting optical flow as pretraining objective) so the comparison is quite weak
- It would be nice to incorporate some baselines on controllable video generation to highlight that AdaWorld is more adaptable than prior works as a result of the learned latent action space.
- The data diversity experiments seem a bit orthogonal to the purpose of the paper and is not really an ablation. The action-agnostic video pretraining is more of an ablation than an actual baseline.

**Questions For Authors:**

Question:
- How does the latent action model avoid collapse, e.g. how do you ensure that the decoder or the forward dynamics model in this case does not just pass f_t through and learn a policy on the demonstration data?
- In Figure 4, what is meant by context-invariant? Does that mean the same latent actions are applied to autoregressively predict the future frames across each of these environments?
- Why do you think optical flow as a condition is not as powerful as image reconstruction?
- Does the average embedding also work in continuous control environments? The highlight example seems to be only in the Procgen discrete action space setting.
- What is the pretraining data for the main set of experiments? Do those use the full OpenX and Retro datasets for pretraining the latent actions?
- What part of the model ensures that the learned latent actions are clustered semantically? Is it the beta-VAE term that controls the amount of disentanglement in thel latent space? Again, is this only application in the discrete domain? Is there any evidence that this holds in continuous environments?

**Relation To Broader Scientific Literature:**

The proposed work is relevant to literature in latent action models and learning from actionless videos including LAPO and Genie among others. The proposed method of learning a latent action space to guide world model generation is sound and a good idea.

**Theoretical Claims:**

There are no theoretical claims and proofs.

---

> ### Author Rebuttal · Authors · 2025-04-01
>
> Thanks for the thoughtful feedback. We answer each question below and will include all results in the revision.
>
> > Compare to state-of-the-art baselines.
>
> **R2-1**: To demonstrate the generality of our method, we use iVideoGPT as a state-of-the-art baseline. iVideoGPT is an action-controlled world model with an autoregressive Transformer architecture. It is pretrained by action-agnostic video prediction and adds a linear projection to learn action control during finetuning. For fair comparison, we implement a variant by conditioning iVideoGPT with our latent actions during pretraining. We resume from the official OpenX checkpoint and do not finetune its tokenizer. After pretraining iVideoGPT and our action-aware variant on OpenX for 10k extra steps, we finetune each model with robot actions for 1k steps on BAIR robot pushing dataset. The result is shown below:
>
> | | PSNR &uarr; | LPIPS &darr; |
> | :--- | :---: | :---: |
> | iVideoGPT | 16.69 | 0.221 |
> | iVideoGPT + AdaWorld | **17.33** | **0.207** |
>
> In our paper, we also compare Genie's latent action setting. The results validate our superior adaptability compared to recent arts.
>
> > Discuss LAPA and PVG. Method is technically not very novel.
>
> **R2-2**: Our key innovation is to incorporate action information into world model pretraining, which significantly enhances its adaptability. To achieve this, we extract continuous latent actions as a scalable condition for our world model. While prior works have studied latent actions, they mainly focus on imitation learning (LAPA, LAPO) and playability (PVG, Genie). As a result, they use discrete latent actions, which struggle to express various actions and fall short in adaptation. As mentioned in Sec. 2.3., our continuous design also enables several unique applications.
>
> > How does the latent action model avoid collapse?
>
> **R2-3**: Our model is less likely to collapse for three main reasons. (1) A large portion of our data is collected randomly. Thus, our model must learn from latent actions. Otherwise, it has no way to predict the next step. (2) Our data contains thousands of environments, making learning a shared action space easier than remembering all behaviors as a decoder policy. (3) The parameter $\beta$ is adjusted to allow sufficient information to pass the bottleneck, ensuring that the latent actions capture meaningful information.
>
> > In Figure 4, what is meant by context-invariant?
>
> **R2-4**: The latent action sequence from the source video is directly applied to the target scene for autoregressive prediction, where the same actions are replicated even when context is totally different.
>
> > Why optical flow as a condition is not as powerful as image reconstruction?
>
> **R2-5**: The dense and uniform optical flow is highly sensitive to spatial and structural misalignment. In contrast, our latent action can adaptively allocate its capacity to represent the most critical actions, making it more robust in recognizing misaligned actions [[LINK](https://icml2025-1014.github.io)].
>
> > Does the average embedding also work in continuous control environments?
>
> **R2-6**: While the average embedding is not directly applicable, AdaWorld still exhibits strong adaptability for continuous action spaces. To verify this, we use nuScenes, an autonomous driving dataset where the vehicle takes continuous displacements at each timestep, as a typical example. During adaptation, we add a two-layer MLP to map actions to the latent action interface. The interface can also be efficiently initialized by finetuning the MLP with minimal action-latent action pairs (3k steps take less than 30 seconds on a single GPU). The result is shown below:
>
> | | PSNR &uarr; | LPIPS &darr; |
> | :--- | :---: | :---: |
> | Act-agnostic | 20.86 | 0.475 |
> | Flow | 20.94 | 0.462 |
> | Discrete | 21.28 | 0.450 |
> | AdaWorld | **21.60** | **0.436** |
>
> We also plot the PSNR curves [[LINK](https://icml2025-1014.github.io)], where AdaWorld adapts more rapidly in all cases. **R2-1** also shows that our method works for another world model with continuous control, while **R4-4** shows that AdaWorld enables better planning for robot arms.
>
> > Pretraining data for main experiments.
>
> **R2-7**: Except for Table 4, all models are pretrained on the data mixture specified in Appendix A.2, including the full OpenX and Retro datasets.
>
> > What part ensures that the latent actions are clustered semantically? Is it the beta-VAE term that controls the amount of disentanglement? Is it only an application in the discrete domain?
>
> **R2-8**: The semantic clustering and disentangling ability result from the low dimensionality of latent actions and the regularization on posterior distributions. The parameter $\beta$ controls the disentanglement of latents [[LINK](https://icml2025-1014.github.io)] (see **R4-1**). This ability also holds for continuous actions. As [[LINK](https://icml2025-1014.github.io)] shows, continuous actions can be effectively disentangled and transferred across contexts.

---

### Official Review · Reviewer_kWaF · 2025-03-15

**Overall Recommendation:** 3

**Summary:**

This paper introduces AdaWorld, a world model learning approach that leverages self-supervised latent action extraction from videos to capture key transitions. It also introduces an autoregressive world model (single-frame SVD) conditioned on these latent actions and historical frames, enabling transfer and learning of new actions.

**Claims And Evidence:**

The paper largely supports its claim regarding action/motion representation learning. However, since it is presented as a world model, it differs from traditional world models where direct user input (e.g., via keyboard or mouse) influences predictions. Instead, this approach requires a reference video to guide actions in the generated video. It would be helpful for the authors to clarify how the model can achieve similar controllability to Genie or OASIS at inference time.

**Essential References Not Discussed:**

A group of motion transfer methods are missing for inference:

Space-Time Diffusion Features for Zero-Shot Text-Driven Motion Transfer, CVPR 2024
MotionClone: Training-Free Motion Cloning for Controllable Video Generation, ICLR 2025

**Experimental Designs Or Analyses:**

For the comparison methods, I suggest that the authors include motion transfer approaches, as referenced earlier. Also consider add user study for action transferring performance.

**Methods And Evaluation Criteria:**

The proposed method makes sense, as learning motion representation through pretraining a latent action autoencoder and applying it to a one-frame denoising diffusion model is a reasonable approach. However, I have some concerns regarding the training data and process (see the section below).

Regarding the evaluation criteria, I am skeptical about the robustness of the ECS metric based on I3D features for assessing action/motion transfer in open-domain scenarios. A more comprehensive user study may be necessary to better evaluate the effectiveness of action/motion transfer.

**Other Comments Or Suggestions:**

None.

**Other Strengths And Weaknesses:**

Strengths: Modeling action or motion using latent representations is a crucial challenge in world models, and this work takes a promising direction by expanding the action vocabulary beyond traditional action-labeled approaches. The proposed method—first learning to extract latent actions and then incorporating them into diffusion models—seems well-motivated and conceptually sound.

Weakness: Please refer to the sections above, as well as the "Questions for Authors" section, for my specific concerns and areas where further clarification is needed.

**Questions For Authors:**

1. In Figure 3, the input appears to be the next frame  f_{t+1}, but this seems to result in a reconstruction rather than a next-frame prediction. Based on the text in Line 248, the last frame in the memory is used as the condition image, which suggests it might be f_t instead?

2. The current approach seems to entangle motion with structural elements, leading to a strong similarity in overall flow between the source and target videos. It would be helpful for the authors to discuss potential limitations of this, particularly in scenarios where the source and target videos are misaligned due to differences in camera poses or character locations. Can the method still function effectively under such conditions?

**Relation To Broader Scientific Literature:**

This paper is related to to diffusion models, world models, and action-driven video generative models.

**Theoretical Claims:**

The theoretical claims appear to be valid. However, I have some concerns about whether the latent action autoencoder effectively captures important actions. Smooth transitions between frames, such as a character simply moving forward, dominate the sequence, whereas sudden transitions, like jumping or performing distinct actions, constitute only a small fraction of the total frames. Yet, during training, it seems that f_t and f_{t+1} are sampled uniformly. This raises the question of whether smooth transitions should be considered "actions" or merely "motion."

---

> ### Author Rebuttal · Authors · 2025-04-01
>
> Thanks for the insightful feedback. We answer each question below and will include all results in the revision.
>
> > The model requires a reference video to guide actions. Clarify how the model can achieve similar controllability to Genie or OASIS at inference time.
>
> **R1-1**: Our world model does not necessitate a reference video for prediction. As shown in Sec. 3.2, our model can be efficiently adapted to various action inputs. After adaptation, our model can be directly controlled by raw actions (e.g., Minecraft actions like OASIS) without using reference videos. Moreover, as mentioned in Sec. 2.3, one can easily obtain a customizable number of control options by clustering latent actions from videos.
>
> > A user study may be necessary to evaluate the effectiveness of action/motion transfer.
>
> **R1-2**: We conduct a user study with the baselines as requested. We follow the same setup in Sec 3.1 and generate 50 video pairs on LIBERO and SSv2, respectively. We then invite four volunteers to judge whether the action is successfully transferred. The success rates are reported below:
>
> | | LIBERO | SSv2 |
> | :--- | :---: | :---: |
> | Act-agnostic | 0% | 1% |
> | Flow | 2% | 10.5% |
> | Discrete | 3.5% | 21.5% |
> | AdaWorld | **70.5%** | **61.5%** |
>
>
> The results indicate that our interface can transfer actions more effectively, especially on LIBERO, where the robot actions are more nuanced.
>
> > Include motion transfer approaches for comparison.
>
> **R1-3**: Thanks for the suggestion. First, we want to emphasize that our paper aims to develop a highly adaptable interface for world models. In contrast, motion transfer methods mainly rely on transferring video-level feature maps, which are not applicable for interactive control and are not suitable for action adaptation. It is also notable that our action transfer ability does not require strictly aligned spatial structures, which is much more flexible than common motion transfer settings (see **R1-7** below).
>
> Due to time limit, we compare the suggested MotionClone using 32 official demos released on its GitHub page. To avoid potential bias, we crop them to square and ensure that the text at the top is excluded. We use AdaWorld to autoregressively predict 16 frames and resize them to MotionClone’s resolution. Since ground truth target videos are not available, we invite four volunteers for a user study. As a result, AdaWorld is preferred 21.09% of time in action transfer accuracy, showing our ability to perform motion transfer tasks (though optimizing video-level motion transfer is not our main purpose).
>
> > Whether the latent action autoencoder effectively captures important actions. During training, smooth transitions between frames dominate the sequence, whereas sudden transitions constitute only a small fraction of the total frames. Whether smooth transitions should be considered "actions" or merely "motion".
>
> **R1-4**: We add more action transfer results to show the capability of our latent action autoencoder [[LINK](https://icml2025-1014.github.io)]. Without any special handling of training data, our model can effectively capture and transfer sudden transitions. Rebalancing the training data may further enhance the learning of these transitions. Note that the latent action autoencoder is encouraged to encode the most critical actions due to the information bottleneck, while minor motions and background changes that can be predicted by the decoder are likely not to be encoded.
>
> > Missing motion transfer references.
>
> **R1-5**: We have included the two references in the related work and will update them accordingly in the revision.
>
> > The input frame in Figure 3.
>
> **R1-6**: Sorry for the confusion. Figure 3 illustrates our training process, where $f_{t+1}$ is the frame to be denoised and is used to generate latent actions. During inference, $f_{t+1}$ is not used and the predictions are made based on past frames. The control interface can take latent actions transferred from other videos, selected from a known set, or raw actions after efficient adaptation. We will clarify this in the revision.
>
> > Can the method still function effectively when the source and target videos are misaligned due to differences in camera poses or character locations?
>
> **R1-7**: Unlike typical motion transfer settings, our method does not require strong spatial alignment to capture actions. As shown in this [[LINK](https://icml2025-1014.github.io)], our latent actions can effectively recognize and transfer actions from various poses and embodiments. Moreover, we want to clarify that the main objective of this work is to develop a generally adaptable world model rather than maximize action representation precision. Although our latent action may not faithfully represent all kinds of actions, it is general enough to serve as a unified interface for pretraining, which significantly improves the adaptability of world models compared to existing training methods. We will add more results in the revision.

---

### Decision · Program_Chairs · 2025-05-01

**Decision:**

Accept (poster)

**Comment:**

This paper introduces AdaWorld, a two-stage training framework designed to learn world models. AdaWorld initially learns a latent action representation using an unsupervised forward prediction objective, and then learns an autoregressive world model that generates future frames by conditioning on the learned latent actions. AdaWorld's effectiveness is demonstrated through evaluation on various downstream tasks, including action transfer between different video contexts and visual planning.

All reviewers agreed on the quality of the paper, and that the paper's claims were well supported by the experimentation. I agree with the reviewers' consensus and therefore support acceptance.